# Direct Training of SNN using Local Zeroth Order Method

**Bhaskar Mukhoty**[1,*]    **Velibor Bojković**[1,*]    **William de Vazelhes**[1]    **Xiaohan Zhao**[3]

**Giulia De Masi**[2,6]    **Huan Xiong**[1,5,†]    **Bin Gu**[1,4,†]

[1] Mohamed bin Zayed University of Artificial Intelligence, UAE
[2] ARRC, Technology Innovation Institute, UAE
[2] Nanjing University of Information Science and Technology, China
[4] School of Artificial Intelligence, Jilin University, China
[5] Harbin Institute of Technology, China
[6] BioRobotics Institute, Sant'Anna School of Advanced Studies, Pisa, Italy

## Abstract

Spiking neural networks are becoming increasingly popular for their low energy requirement in real-world tasks with accuracy comparable to traditional ANNs. SNN training algorithms face the loss of gradient information and non-differentiability due to the Heaviside function in minimizing the model loss over model parameters. To circumvent this problem, the surrogate method employs a differentiable approximation of the Heaviside function in the backward pass, while the forward pass continues to use the Heaviside as the spiking function. We propose to use the zeroth-order technique at the local or neuron level in training SNNs, motivated by its regularizing and potential energy-efficient effects and establish a theoretical connection between it and the existing surrogate methods. We perform experimental validation of the technique on standard static datasets (CIFAR-10, CIFAR-100, ImageNet-100) and neuromorphic datasets (DVS-CIFAR-10, DVS-Gesture, N-Caltech-101, NCARS) and obtain results that offer improvement over the state-of-the-art results. The proposed method also lends itself to efficient implementations of the back-propagation method, which could provide 3-4 times overall speedup in training time. The code is available at `https://github.com/BhaskarMukhoty/LocalZO`.

## 1 Introduction

Biological neural networks are known to be significantly more energy efficient than their artificial avatars - the artificial neural networks (ANN). Unlike ANNs, biological neurons use spike trains to communicate and process information asynchronously. [27] To closely emulate biological neurons, spiking neural networks (SNN) use binary activation to send information to the neighbouring neurons when the membrane potential exceeds membrane threshold. The event-driven binary activation simplifies the accumulation of input potential and reduces the computation burden when the spikes are sparse. Specialized neuromorphic hardware [11] is designed to carry out such event-driven and sparse computations in an energy-efficient way [32, 20].

---

[*]First co-author

[†]Correspondence to Huan Xiong (huan.xiong.math@gmail.com), and Bin Gu (jsgubin@gmail.com).

37th Conference on Neural Information Processing Systems (NeurIPS 2023).

There are broadly three categories of training SNNs: ANN-to-SNN conversion, unsupervised and supervised. The first one is based on the principle that parameters for SNN can be inferred from the corresponding ANN architecture [6, 15, 5]. Although training SNNs through this method achieves performance comparable to ANNs, it suffers from the long latency needed in SNNs to emulate the corresponding ANN or from retraining of ANNs required to achieve near lossless conversion [10]. The unsupervised training is biologically inspired and uses local learning to adjust the SNN parameters [14]. Although it is the most energy-efficient one among the three as it is implementable on neuromorphic chips [11], it still lags in its performance compared to ANN-to-SNN conversion and supervised training.

Finally, supervised training is a method of direct training of SNNs by using back-propagation (through time). As such, it faces two main challenges. The first is due to the nature of SNNs, or more precisely, due to the Heaviside activation of neurons (applied to the difference between the membrane potential and threshold). As the derivative of the Heaviside function is zero, except at zero where it is not defined, back-propagation does not convey any information for the SNN to learn [16]. One of the most popular ways to circumvent this drawback is to use surrogate methods, where a derivative of a surrogate function is used in the backward pass during training. Due to their simplicity, surrogate methods have been widely used and have seen tremendous success in various supervised learning tasks [36, 28]. However, large and complex network architectures, the time-recursive nature of SNNs, and the fact that the training is oblivious of the sparsity of spikes in SNNs make surrogate methods quite time and energy-consuming.

Regarding regularization or energy efficiency during direct training of SNNs, only a few methods have been proposed addressing these topics together or separately, most of which deal with the forward propagation in SNNs. For example, [1] uses stochastic neurons to increase energy efficiency during inference. More recently, [42] uses regularization during the training to increase the sparsity of spikes, reducing the computational burden and energy consumption. Further, [7] performs the forward pass on a neuromorphic chip, while the backward pass is performed on a standard GPU. Although these methods improve the SNN models' performance, they do not significantly reduce the computational burden or provide the potential to do so. On the other hand, [31] introduces a threshold for surrogate gradients (or suggests using only a surrogate with bounded support). However, introducing gradient thresholds has the drawback of limiting the full potential of surrogates during training.

This paper proposes a direct training method for SNNs based on the zeroth order technique. We apply it locally, at the neuronal level - hence dubbed Local Zeroth Order (LOCALZO) - with twofold benefits: regularization, which comes as a side-effect of the introduced randomness that is naturally associated with this technique, as well as a threshold for gradient backpropagation in the style of [31] which translates to potential energy-efficient training when properly implemented.

We summarize the main contributions of the paper as follows:

- We introduce zeroth order techniques in SNN training at a local level. We provide extensive theoretical properties of the method, relating it to the surrogate gradients via the internal distributions used in LOCALZO.

- We experimentally demonstrate the main properties of LOCALZO: its superior performance compared to baselines when it comes to generalizations, its ability to simulate arbitrary surrogates as well as its property to speed up the training process, which translates to energy-efficient training.

## 2 Background

### 2.1 Spiking neuron dynamics

An SNN consists of Leaky Integrate and Fire neurons (LIF) governed by differential equations in continuous time [19]. They are generally approximated by discrete dynamics given in the form of

recurrence equations,

$$u_i^{(l)}[t] = \beta u_i^{(l)}[t-1] + \sum_j w_{ij} x_j^{(l-1)}[t] - x_i^{(l)}[t-1]u_{th},$$

$$x_i^{(l)}[t] = h(u_i^{(l)}[t] - u_{th}) = \begin{cases} 1 & \text{if } u_i^{(l)}[t] > u_{th} \\ 0 & \text{otherwise,} \end{cases} \tag{1}$$

where $u_i^{(l)}[t]$ denote the membrane potential of $i$-th neuron in the layer $l$ at time-step (discrete) $t$, which recurrently depends upon its previous potential (with scaling factor $\beta < 1$) and spikes $x_j^{(l-1)}[t]$ received from the neurons of previous layers weighted by $w_{ij}$. The neuron generates binary spike $x_i^{(l)}[t]$ whenever the membrane potential exceeds threshold $u_{th}$, represented by the Heaviside function $h$, followed by a reset effect on the membrane potential.

To implement the back-propagation of training loss through the network, one must obtain a derivative of the spike function, which poses a significant challenge in its original form represented as:

$$\frac{dx_i[t]}{du} = \begin{cases} \infty & \text{if } u_i^{(l)}[t] = u_{th} \\ 0 & \text{otherwise.} \end{cases} \tag{2}$$

where we denote $u := u_i^{(l)}[t] - u_{th}$. To avoid the entire gradient becoming zero, known as the dead neuron problem, the surrogate gradient method (referred to as SURROGATE) redefines the derivative using a surrogate:

$$\frac{dx_i[t]}{du} := g(u) \tag{3}$$

Here, the function $g(u)$ can be, for example, the derivative of the Sigmoid function (see section 4.4), but in general, one takes a scaled probability density function as a surrogate (see Section 4.2 for more details).

## 2.2 Motivation

Classically, the purpose of dropout in ANNs is to prevent a complex and powerful network from over-fitting the training data, which consequently implies better generalization properties [3]. In the forward pass, one usually assigns to each neuron in a targeted layer a probability of being "switched-off" during both forward and backward passes, and this probability does not change during the training. Moreover, the "activity" of the neuron, however we may define it, does not affect whether the neuron will be switched on or off.

Our motivation comes along these lines: how to introduce a dropout-like regularizing effect in the training of SNNs, but keeping in mind the temporal dimension of the data, as well as the neuron activity at that particular moment (heuristically, a more active neuron would be kept "on" with a high probability (randomness of the dropout) while a less active one would be "switched off", again with high probability, in a sense to be made precise shortly). Generally speaking, our idea consists of the following two steps: 1) For each spiking neuron of our SNN network, measure how active the neuron is in the forward pass at each time step $t$. Here, we define the activity based on how far the current membrane potential of the neuron $u[t]$ is from the firing threshold $u_{th}$ (this idea comes from [31]). However, unlike in [31] where the sole distance is the determining factor, we introduce the effect of randomness via a fixed PDF, say $\lambda$, sample $z$ from it and say the neuron is active at time $t$ if $|u[t] - u_{th}| < c|z|$, where $c$ is some upfront fixed constant. 2) In the backward pass at the time $t$, if the neuron is dubbed active, we will apply some surrogate function $g(u[t] - u_{th})$; otherwise, we will take the surrogate to be 0 (hence, switching off the propagation of gradients through the neuron in the latter case).

Having said this, we ask ourselves the final question: can we have a systematic way of choosing functions $\lambda$ and $g$ so that the expected surrogate we use (with respect to $\lambda$) equals the one we chose upfront? A simple yet elegant solution that satisfies all of the above comes with zeroth order methods.

Zeroth order technique is a popular gradient-free method [26], well studied in neural networks literature. To briefly introduce it, we consider a function $f : \mathbb{R}^d \to \mathbb{R}$, that we intend to minimize

using gradient descent, for which the gradient may not be available or even undefined. The zeroth-order method estimates the gradients using function outputs: given a scalar $\delta > 0$, the 2-point ZO is defined as

$$G^2(\mathbf{w}; \mathbf{z}, \delta) = \phi(d) \frac{f(\mathbf{w} + \delta\mathbf{z}) - f(\mathbf{w} - \delta\mathbf{z})}{2\delta} \mathbf{z} \tag{4}$$

where, $\mathbf{z} \sim \lambda$ is a random direction with $\mathbb{E}_{z \sim \lambda}[\|\mathbf{z}\|^2] = 1$ and $\phi(d)$ is a dimension dependent factor, with $d$ being the dimension. However, to approximate the full gradient of $f$ up to a constant squared error, we need an average of $O(d)$ samples of $G^2$, which becomes computationally challenging when $d$ is large, such as the number of learnable parameters of the neural network. Though well studied in the literature, properties of 2-point ZO are known only for the continuous functions [29, 4]. In the present context, we will apply it locally to the Heaviside function that produces the outputs of spiking neurons, and we justify this by providing the necessary theoretical background for doing so.

## 3 The LOCALZO algorithm

Applying ZO on a global scale is challenging due to the large dimensionality of neural networks[25]. Since the non-differentiability of SNN is introduced by the Heaviside function at the neuronal level, we apply the 2-point ZO method on $h : \mathbb{R} \to \{0, 1\}$ itself,

$$G^2(u; z, \delta) = \frac{h(u + z\delta) - h(u - z\delta)}{2\delta} z = \begin{cases} 0, & |u| > |z|\delta \\ \frac{|z|}{2\delta}, & |u| < |z|\delta \end{cases} \tag{5}$$

where $u = u_i^{(l)}[t] - u_{th}$ and $z$ is sampled from some distribution $\lambda$. We may average the 2-point ZO gradient over a few samples $z_k$ so that the LOCALZO derivative of the spike function is defined as:

$$\frac{dx_i[t]}{dt} := \frac{1}{m} \sum_{k=1}^{m} G^2(u; z_k, \delta) \tag{6}$$

where, the number of samples, $m$, is a hyper-parameter to the LOCALZO method. We implement this at the neuronal level of the back-propagation routine, where the forward pass uses the Heaviside function, and the backward pass uses equation (6). Note that the gradient $\frac{dx_i[t]}{dt}$ being non-zero naturally determines the active neurons of the backward pass (as was discussed in Section 2.2), which can be inferred from the forward pass through the neuron. Algorithm 1 gives an abstract representation of the process at a neuronal level, which hints that the backward call is redundant when the neuron has a zero gradient.

## 4 Theoretical Properties of LOCALZO

### 4.1 General ZO function

For the theoretical results around LOCALZO, we consider a more general function than what was suggested by eqn. 5, in the form

$$G^2(u; z, \delta) = \begin{cases} 0, & |u| > |z|\delta \\ \frac{|z|^\alpha}{2\delta}, & |u| \le |z|\delta, \end{cases} \tag{7}$$

where the new constant $\alpha$ is an integer different from 0, while $\delta$ is a positive real number (so, for example, setting $\alpha = 1$ in (7), we obtain (5)).

The integer $\alpha$ is somewhat a normalizing constant, which allows obtaining different surrogates as the expectation of function $G^2(u; z, \delta)$ when $z$ is sampled from suitable distributions.

---

**Algorithm 1** LOCALZO

**Forward**
**Require:** $u := u_i^{(l)}[t] - u_{th}$, dist. $\lambda$, const. $\delta, m$
    sample $z_1, z_2, \cdots z_m \sim \lambda$
    $grad \leftarrow \frac{1}{m} \sum_{k=1}^{m} \mathbb{I}(|u| < \delta|z_k|) \frac{|z_k|}{2\delta}$
    **if** $grad \neq 0$ **then**
        SaveForBackward($grad$)
    **end if**
    **return** $\mathbb{I}(u > 0)$

---

**Backward** {Invoked if grad is non-zero}
**Require:** gradient from chain rule: $grad\_input$
    **return** $grad\_input * grad$

---

In practice, taking $\alpha = \pm 1$ will suffice to account for most of the surrogates found in the literature. The role of $\delta$ is somewhat different, as it controls the "shape" of the surrogate (narrowing it and stretching around zero). The role of each constant will be more evident from what follows (see section 4.4).

## 4.2 Surrogate functions

**Definition 4.1.** We say that a function $g : \mathbb{R} \to \mathbb{R}_{\geq 0}$ is a surrogate function (gradient surrogate) if it is even, non-decreasing on the interval $(-\infty, 0)$ and $c := \int_{-\infty}^{\infty} g(z)dz < \infty$.

Note that the integral $\int_{-\infty}^{\infty} g(z)dz$ is convergent (as $g(z)$ is non-negative), but possibly can be $\infty$ and the last condition means that the function $\frac{1}{c}g(t)$ is a probability density function. The first two conditions, that is, requirements for the function to be even and non-decreasing, are not essential but rather practical and consistent with examples from SNN literature.

Note that the function $G : \mathbb{R} \to [0, 1]$, defined as $G(t) := \frac{1}{c} \int_{-\infty}^{t} g(z)dz$ is the corresponding cumulative distribution function (for PDF $\frac{1}{c}g(t)$). Moreover, it is not difficult to see that its graph is "symmetric" around point $(0, \frac{1}{2})$ (or in more precise terms, $G(t) = 1 - G(-t)$), hence $G(t)$ can be seen as an approximation of Heaviside function $h(t)$. Then, its derivative $\frac{d}{dt}G(t) = \frac{1}{c}g(t)$ can serve as an approximation of the "derivative" of $h(t)$, or in other words, as its surrogate, which somewhat justifies the terminology.

Finally, one may note that "true" surrogates would correspond to those functions $g$ for which $c = 1$. However, the reason we allow $c$ to be different from 1 is again practical and simplifies the derivation of the results that follow. We note once again that allowing general $c$ is in consistency with examples used in the literature.

## 4.3 Surrogates and ZO

To be in line with classic results around the ZO method and gradient approximation of functions, we pose ourselves two fundamental questions: What sort of functions in variable $u$ can be obtained as the expectation of $G^2(u; z, \delta)$ when $z$ is sampled from a suitable distribution $\lambda$, and, given some function $g(u)$, can we find a distribution $\lambda$ such that we obtain $g(u)$ in the expectation when $z$ is sampled from $\lambda$?

Two theorems that follow answer these questions and are the core of this section. The main player in both of the questions is the expected value of $G^2(u; z, \delta)$, so we start by analyzing it more precisely. Let $\lambda$ be a distribution, $\lambda(t)$ its PDF for which we assume that it is even and that $\int_0^{\infty} z^\alpha \lambda(z)dz < \infty$. Then, we may write

$$\mathbb{E}_{z \sim \lambda}[G^2(u; z, \delta)] = \int_{-\infty}^{\infty} G^2(u; z, \delta)\lambda(z)dz = \int_{|u| \leq |z|\delta} \frac{|z|^\alpha}{2\delta}\lambda(z)dz = \frac{1}{\delta} \int_{\frac{|u|}{\delta}}^{\infty} z^\alpha \lambda(z)dz. \quad (8)$$

It becomes apparent from eqn. (8) that $\mathbb{E}_{z \sim \lambda}[G^2(u; z, \delta)]$ has some properties of surrogate functions (it is even and non-decreasing on $\mathbb{R}_{<0}$). The proofs of the following results are detailed in the appendix A.

**Lemma 1.** *Assume further that $\int_0^{\infty} z^{\alpha+1}\lambda(z)dz < \infty$. Then, $\mathbb{E}_{z \sim \lambda}[G^2(u; z, \delta)]$ is a surrogate function.*

**Theorem 2.** *Let $\lambda$ be a distribution and $\lambda(t)$ its corresponding PDF. Assume that integrals $\int_0^{\infty} t^\alpha \lambda(t)dt$ and $\int_0^{\infty} t^{\alpha+1}\lambda(t)dt$ exist and are finite. Let further $\tilde{\lambda}$ be the distribution with corresponding PDF function*

$$\tilde{\lambda}(z) = \frac{1}{c} \int_{|z|}^{\infty} t^\alpha \lambda(t)dt,$$

*where $c$ is the scaling constant (such that $\int_{-\infty}^{\infty} \tilde{\lambda}(z)dz = 1$). Then,*

$$\mathbb{E}_{z \sim \lambda}[G^2(u; z, \delta)] = \frac{d}{du}\mathbb{E}_{z \sim \tilde{\lambda}}[c\,h(u + \delta z)].$$

For our next result, which answers the second question we asked at the beginning of this section, note that a surrogate function is differentiable almost everywhere, which follows from the Lebesgue theorem on the differentiability of monotone functions. So, taking derivatives here is understood in an "almost everywhere" sense.

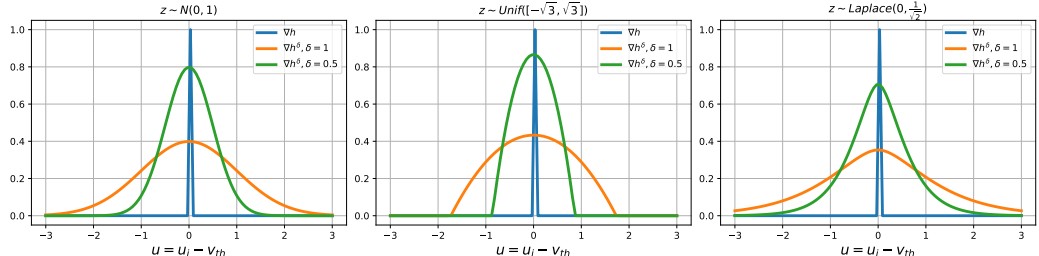

Figure 1: The figure shows the expected surrogates derived in section 4.4 as $z$ is sampled from Normal$(0, 1)$, Unif$([\sqrt{3}, \sqrt{3}])$ and Laplace$(0, \frac{1}{\sqrt{2}})$ respectively. Each figure shows the surrogates corresponding to $\delta \to 0$, $\delta = 0.5$ and 1. The surrogates are supplied to SPARSEGRAD methods for a fair comparison with LOCALZO as the latter uses respective distributions to sample $z$.

**Theorem 3.** *Let $g(u)$ be a surrogate function. Suppose further that $c = -2\delta^2 \int_0^\infty \frac{1}{z^\alpha} g'(z\delta) dz < \infty$ and put $\lambda(z) = -\frac{\delta^2}{cz^\alpha} g'(z\delta)$ (so that $\lambda(z)$ is a PDF). Then,*

$$c \, \mathbb{E}_{z \sim \lambda}[G^2(u; z, \delta)] = \mathbb{E}_{z \sim \lambda}[c \, G^2(u; z, \delta)] = g(u).$$

### 4.4 Application of Theorem 2 and 3

Next, we spell out the results of Theorem 2 applied to some standard distributions, with $\alpha = 1$. For clarity, all the distributions' parameters are chosen so that the scaling constant of the resulting surrogate is 1. One may consult Figure 1 for the visual representation of the results, while the details are provided in the appendix A.1. Recall that the standard normal distribution $N(0, 1)$ has PDF of the form $\frac{1}{\sqrt{2\pi}} \exp(-\frac{z^2}{2})$. Consequently, it is straightforward to obtain

$$\mathbb{E}_{z \sim \lambda}[G^2(u; z, \delta)] = \frac{1}{\sqrt{2\pi}} \int_{-\infty}^\infty \frac{|z|}{2\delta} \exp(-\frac{z^2}{2}) dz = \frac{1}{\delta\sqrt{2\pi}} \exp(-\frac{u^2}{2\delta^2}). \tag{9}$$

In appendix, A.1, we further derive surrogates when $z$ is sampled from Uniform and Laplace distribution.

We recall that Theorem 3 provides a way to derive distributions for arbitrary surrogate functions ( that satisfy the conditions of the theorem). Consider the Sigmoid surrogate function, where the differentiable Sigmoid function approximates the Heaviside [43]. The corresponding surrogate gradient is given by,

$$\frac{dx}{du} = \frac{d}{du} \frac{1}{1 + \exp(-ku)} = \frac{k \exp(-ku)}{(1 + \exp(-ku))^2} =: g(u)$$

Observe that $g(u)$ satisfies our definition of a surrogate ($g(u)$ being even, non-decreasing on $(-\infty, 0)$ and $\int_{-\infty}^\infty g(u) du = 1 < \infty$). Thus, according to Theorem 3, we have $c = -2\delta^2 \int_0^\infty \frac{g'(t\delta)}{t} dt = \frac{\delta^2 k^2}{a^2}$ where, $a := \sqrt{\frac{1}{0.4262}}$. The corresponding PDF is given by

$$\lambda(z) = -\frac{\delta^2}{c} \frac{g'(\delta t)}{z} = a^2 \frac{\exp(-k\delta z)(1 - \exp(-k\delta z))}{z(1 + \exp(-k\delta z))^3} \tag{10}$$

Observe that the temperature parameter $k$ comes from the surrogate to be simulated, while $\delta$ is used by LOCALZO. Appendix A.2 provides calculations and distribution corresponding to the popular Fast-sigmoid surrogate, followed by a description of the inverse sampling method that can be used to simulate sampling for arbitrary distributions using the uniform distribution.

### 4.5 Expected back-propagation threshold for LOCALZO

To study the energy efficiency of the LOCALZO method when training SNNs, we compute the expected threshold $\widetilde{B}_{th}$ for the activity of the neurons, i.e. the expectation of the quantity $|z|\delta$ when $z$ is sampled from a distribution $\lambda$. It is used in the experimental section when comparing our method with the alternative energy-efficient method [31]. The expected threshold values are presented in Table 1 ($m$ denotes the number of samples used in (6)), while the details of the derivations can be found in A.3.

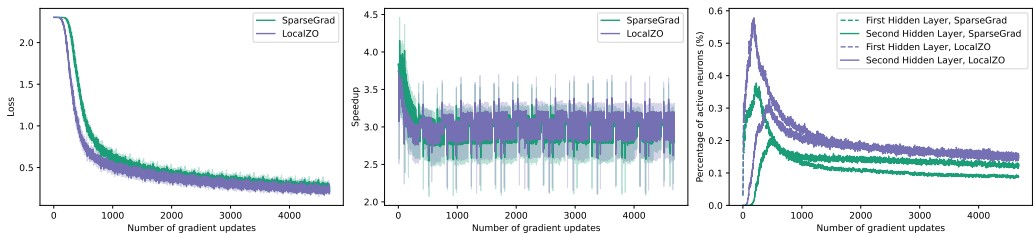

Figure 2: We plot training loss, overall speedup, and percentage of active neurons for the Sigmoid surrogate, as reported in Table 3. The LOCALZO algorithm converges faster than the SPARSEGRAD method while having a similar overall speedup. The percentage of active neurons being less than 0.6% explains the reduced computational requirement, which translates to backward speedup.

# 5 Experiments

## 5.1 General Performance of LOCALZO

First, we evaluate the generalization performance of LOCALZO as a substitute for the surrogate method on standard static image datasets such as CIFAR-10, CIFAR-100[22], ImageNet-100[12] and neuromorphic datasets such as DVS-CIFAR-10[24], DVS-Gesture[2], N-Caltech-101[30], N-CARS[37]. More specifically, by substituting surrogate gradients with gradients computed with our method, we com-

Table 1: The expected back-propagation thresholds

| $z \sim \lambda$ | | $\tilde{B}_{th}/\delta$ | |
|---|---|---|---|
| $\lambda$ | $F_{|z_k|}(x)$ | $m = 1$ | $m = 5$ |
| Normal$(0, 1)$ | $\text{erf}(\frac{x}{\sqrt{2}})$ | 0.798 | 1.569 |
| Unif$([\sqrt{3}, \sqrt{3}])$ | $\frac{x}{\sqrt{3}}$ | 0.866 | 1.443 |
| Laplace$(0, \frac{1}{\sqrt{2}})$ | $1 - e^{-\sqrt{2}x}$ | 0.707 | 1.615 |

bined LOCALZO with contemporary state-of-the-art methods in direct training of SNNs, such as tDBN [45] and TET [13]. We opted for these two techniques due to their high performance and different natures (the former is a batch normalization technique, while the latter introduces auxiliary loss in training). We refer to the combined methods as LocalZO+tDBN and LocalZO+TET, respectively. Following the recent results, we choose ResNet-19[45] architecture for CIFAR datasets, SEW-ResNet-34[17] for ImageNet-100, and VGGSNN[13] architecture for the neuromorphic datasets. Table2 summarizes these results, where LOCALZO is implemented with $m = 5, \delta = 0.5$, except for DVS-CIFAR-10 where we have used $m = 1$. Table 7 reports the detailed training hyper-parameters for each dataset.

**Results on Static Datasets**: The static datasets CIFAR-10, CIFAR-100, and ImageNet-100, have the number of classes mentioned in the dataset name, while each class respectively have (5000, 1000), (500, 100), and (1300, 50) train and test images. We use constant encoding to supply the images to the SNN network, with standard latencies as mentioned in Table 2. We train separate models for different latencies and report the test results respectively. The results of CIFAR datasets are reported with cutout augmentation following TET[13], while for ImageNet-100, we use standard data augmentation (random resized crop, random horizontal flip), without and with the ImageNet Policy[9]. We note that in conjunction with LOCALZO, both tDBN and TET improve their performance significantly for all the latencies. For example, in CIFAR-10, LOCALZO improves TET between $0.9 - 1\%$ for different latencies, while for CIFAR-100, it improves TET by $2.5 - 3.5\%$. The ImageNet-100 results reported in Table 2 were further enhanced to 83.33% with $m = 20$ and batch size 72.

**Results on Neuromorphic Datasets**: Events of the neuromorphic datasets are collected into event frames of dimension $(2 \times H \times W)$ where H and W stand for the height and width of the frame and are resized to $(2 \times 48 \times 48)$. The temporal events are collected into a fixed number (10) of frames (a.k.a. bins), treated as the effective temporal dimension for the SNN. The experiments are reported without and with standard data augmentation. For DVS-CIFAR-10, the TET result is re-computed to avoid an obscure frame preparation step, which is replaced by an open-source routine. We obtain results superior to the state-of-the-art for DVS-CIFAR-10[13] and DVS-Gesture[17]. For N-Caltech-101 and NCARS, the improvements are 1.2% and 2.4%, respectively, compared to the state-of-the-art[18, 35].

Table 2: Comparison with the existing methods show that LOCALZO improves the accuracy of existing direct training algorithms. For the existing methods, we compare the performance with the results reported in respective literatures. For the rows with two accuracies reported, the second one is for training with additional augmentation.

| Dataset | Methods | Architecture | Simulation Length | Accuracy |
|---|---|---|---|---|
| CIFAR10 | Hybrid training[34] | ResNet-20 | 250 | 92.22 |
| | Diet-SNN[33] | ResNet-20 | 10 | 92.54 |
| | STBP[38] | CIFARNet | 12 | 89.83 |
| | STBP NeuNorm[39] | CIFARNet | 12 | 90.53 |
| | TSSL-BP[44] | CIFARNet | 5 | 91.41 |
| | tDBN[45] | ResNet-19 | 6 | 93.16 |
| | | | 4 | 92.92 |
| | | | 2 | 92.34 |
| | **LOCALZO +tDBN** | ResNet-19 | 6 | 95.07 |
| | | | 4 | 94.89 |
| | | | 2 | 94.65 |
| | TET[13] | ResNet-19 | 6 | 94.50 |
| | | | 4 | 94.44 |
| | | | 2 | 94.16 |
| | **LOCALZO +TET** | ResNet-19 | 6 | **95.56** |
| | | | 4 | **95.3** |
| | | | 2 | **95.03** |
| CIFAR100 | Hybrid training[34] | VGG-11 | 125 | 67.87 |
| | Diet-SNN[33] | ResNet-20 | 5 | 64.07 |
| | tDBN[45] | ResNet-19 | 6 | 71.12 |
| | | | 4 | 70.86 |
| | | | 2 | 69.41 |
| | **LOCALZO +tDBN** | ResNet-19 | 6 | 73.74 |
| | | | 4 | 74.13 |
| | | | 2 | 72.78 |
| | TET[13] | ResNet-19 | 6 | 74.72 |
| | | | 4 | 74.47 |
| | | | 2 | 72.87 |
| | **LOCALZO +TET** | ResNet-19 | 6 | **77.25** |
| | | | 4 | **76.89** |
| | | | 2 | **76.36** |
| ImageNet-100 | EfficientLIF-Net[21] | ResNet-19 | 5 | 79.44 |
| | **LOCALZO +TET** | SEW-Resnet34 | 4 | 78.58, **81.56**[‡] |
| DVS-CIFAR10 | tdBN[45] | ResNet-19 | 10 | 67.8 |
| | Streaming Rollout [23] | DenseNet | 10 | 66.8 |
| | Conv3D[40] | LIAF-Net | 10 | 71.70 |
| | LIAF[40] | LIAF-Net | 10 | 70.40 |
| | TET[13] | VGGSNN | 10 | 74.89[⋆], 81.45[⋆] |
| | **LOCALZO +tDBN** | VGGSNN | 10 | 72.6, 79.37 |
| | **LOCALZO +TET** | VGGSNN | 10 | **75.62, 81.87** |
| N-Caltech-101 | AEGNN[35] | GNN | - | 66.8 |
| | EST[18] | ResNet-34[†] | 9 | 81.7 |
| | **LOCALZO +tDBN** | VGGSNN | 10 | 74.65, 79.05 |
| | **LOCALZO +TET** | VGGSNN | 10 | **79.86, 82.99** |
| N-CARS | AEGNN[35] | GNN | - | 94.5 |
| | EST[18] | ResNet-34[†] | 9 | 92.5 |
| | **LOCALZO +tDBN** | VGGSNN | 10 | 95.96, 95.68 |
| | **LOCALZO +TET** | VGGSNN | 10 | **96.78, 96.96** |
| DVS-Gesture | SEW[17] | SEW-Resnet | 16 | 97.92 |
| | **LOCALZO +TET** | VGGSNN | 10 | 98.04, **98.43** |

[⋆] our implementation, [†] pre-trained with ImageNet, [‡] 83.33 % with $m = 20$

## 5.2 Performance on Energy Efficient Implementation

In the energy-efficient implementation of the back-propagation [31], the optimization of the network weights takes place in a layer-wise fashion through the unrolling of recurrence of equation (1) w.r.t time. As the active neurons of each layer for every time step are inferred from the forward pass, gradients of only active neurons are required to be saved for the backward pass, hence saving the computation requirement of the backward pass. One may refer to [31] for further details of this implementation framework. To compare, we supply SPARSEGRAD method the surrogate approximated by LOCALZO, as per section 4.4. The SPARSEGRAD algorithm also requires a back-propagation threshold parameter, $B_{th}$, to control the number of active neurons participating in the back-propagation. We supply it the expected back-propagation threshold $\tilde{B}_{th}$ of LOCALZO as obtained in sections 4.5. We follow the same experimental setting as in [31] for a fair comparison. We use a fully connected LIF neural network with two hidden layers of 200 neurons each and input and output layers. We train every model for 20 epochs and report the average training and test accuracies computed over five trials. We compute the speedup of SPARSEGRAD and LOCALZO, with respect to the full surrogate without truncation, that uses standard back-propagation. The backward speedup (Back.) captures the number of times the backward pass of a gradient update is faster, while the overall speedup (Over.) considers the total time for the forward and the backward pass and then computes the speedup. The speedup reported is averaged over all the gradient updates and the experimental trials.

We compare the performance of the algorithms on three datasets: 1) Neuromorphic-MNIST (NMNIST) [30], which consists of static images of handwritten digits (between 0 and 9) converted to temporal spiking data using visual neuromorphic sensors; 2) Spiking Heidelberg Digits (SHD) [8], a neuromorphic audio dataset consisting of spoken digits (between 0 and 9) in English and German language, totalling 20 classes. To challenge the generalizability of the learning task, 81% of test inputs of this dataset are new voice samples, which are not present in the training data; 3) Fashion-MNIST (FMNIST) [41] dataset is converted using temporal encoding to convert static gray-scale images based on the principle that each input neuron spikes only once, and a higher intensity spike results in an earlier spike.

Table 3 provides a comparison of the algorithms, using surrogates corresponding to the Normal and Sigmoid, with $\delta = 0.05$ and $m = 1$. For the normal distribution, we supply SPARSEGRAD algorithm the back-propagation threshold $\tilde{B}_{th}$ obtained in Table 1. In the section 4.4, we derived distributions corresponding to the Sigmoid surrogate. We use inverse transform sampling (see A.2.3), and take the temperature parameter $k = a/\delta \approx 30.63$ so that $c = \frac{\delta^2 k^2}{a^2} = 1$ and supply SPARSEGRAD method the corresponding back-propagation threshold, $\tilde{B}_{th} = 0.766\delta$. The LOCALZO method offers better test accuracies than SPARSEGRAD, with a slight compromise in speedup due to the sampling of random variable $z$. The difference between training and test accuracies for the SHD dataset can be attributed to the unseen voice samples in the test data[8].

Figure 2 shows the training loss, overall speedup, and percentage of active

Table 3: Performance on NMNIST, SHD and FMNIST

| Method | Train | Test | Back. | Over. |
|---|---|---|---|---|
| **NMNIST** | $z \sim \text{Normal}(0,1), \delta = 0.05, m = 1$ | | | |
| SPARSEGRAD | $93.26 \pm 0.31$ | $91.86 \pm 0.29$ | 99.57 | 3.38 |
| LOCALZO | $94.38 \pm 0.12$ | $93.29 \pm 0.08$ | 92.27 | 3.34 |
| | Sigmoid, $\delta = 0.05, k \approx 30.63, m = 1$ | | | |
| SPARSEGRAD | $92.96 \pm 0.26$ | $91.04 \pm 0.32$ | 87.45 | 3.00 |
| LOCALZO | $93.98 \pm 0.08$ | $92.97 \pm 0.05$ | 83.54 | 3.02 |
| **SHD** | $z \sim \text{Normal}(0,1), \delta = 0.05, m = 1$ | | | |
| SPARSEGRAD | $92.03 \pm 0.79$ | $74.73 \pm 0.73$ | 143.7 | 4.83 |
| LOCALZO | $91.77 \pm 0.27$ | $76.55 \pm 0.93$ | 142.8 | 4.75 |
| | Sigmoid, $\delta = 0.05, k \approx 30.63, m = 1$ | | | |
| SPARSEGRAD | $92.19 \pm 0.41$ | $75.80 \pm 0.97$ | 140.8 | 4.46 |
| LOCALZO | $91.96 \pm 0.11$ | $76.97 \pm 0.40$ | 133.6 | 4.36 |
| **FMNIST** | $z \sim \text{Normal}(0,1), \delta = 0.05, m = 1$ | | | |
| SPARSEGRAD | $81.91 \pm 0.10$ | $80.28 \pm 0.11$ | 15.74 | 1.97 |
| LOCALZO | $83.83 \pm 0.07$ | $81.79 \pm 0.06$ | 15.49 | 1.88 |
| | Sigmoid, $\delta = 0.05, k \approx 30.63, m = 1$ | | | |
| SPARSEGRAD | $81.60 \pm 0.11$ | $80.02 \pm 0.08$ | 12.12 | 1.65 |
| LOCALZO | $83.39 \pm 0.10$ | $81.76 \pm 0.10$ | 12.50 | 1.57 |

neurons after each gradient step for the Sigmoid surrogate. The sparseness of active neurons (under 0.6%) explains the reduced computational requirement that translates to the speedup.

We further implement LOCALZO with $\delta = 0.5, z \sim$ Normal$(0, 1)$ to train a CNN architecture (Input- 16C5-BN-LIF-MP2-32C5-BN-LIF-MP2-800FC-10) and compare it with the corresponding surrogate gradient algorithm. Figure 3 shows the corresponding sparsity of the methods by plotting the number of zero elements at a neuronal level. The plot suggests that during the training, LOCALZO exhibits higher sparsity gradients than the surrogate method.

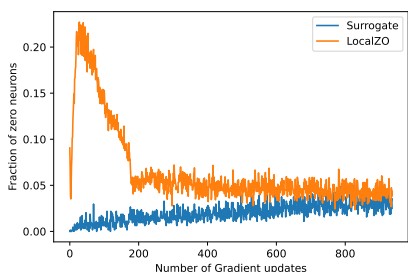

**Ablation study:** Table 4 further shows the test accuracy of the LOCALZO method and overall speed-up for a wide range of values of $m$ with $z \sim$ Normal$(0, 1)$. Like Table 3, the experiments are repeated five times and mean test accuracy is reported along with standard deviation (Std.).

Figure 3: Comparison of gradient sparsity in CNN architecture

In general, by increasing $m$, the method approximates the surrogate better, still offers the regularizing effect and potentially improves the generalization, but also requires more computation. Larger $m$ leads to more non-zero gradients at the neuronal level in the backward pass, reducing overall speed-up. On the other hand, smaller $m$ introduces higher randomness (less "controlled"), still yielding regularization, which helps obtain better generalization, as well as potential speed-up. In conclusion, $m$ should be treated as a hyper-parameter, its value depending on the training setting itself. In our experiments, we chose $m = 1$ or $5$ for most of the experiments, as a proof of concept, but also because it offers a nice balance between the speed-up and performance.

Table 4: Trade-off of accuracy vs. speedup with hyper-parameter $m$

| m | 1 | 3 | 5 | 7 | 10 | 20 | 100 |
|---|---|---|---|---|---|---|---|
| **NMNIST** | | | | | | | |
| Accuracy | 93.29 | 93.61 | 93.69 | 93.66 | 93.76 | 93.67 | 93.81 |
| Std. | 0.08 | 0.15 | 0.17 | 0.13 | 0.14 | 0.08 | 0.14 |
| Over. | 3.33 | 3.28 | 3.22 | 3.16 | 3.06 | 2.82 | 1.59 |
| **SHD** | | | | | | | |
| Accuracy | 76.55 | 76.55 | 76.50 | 75.49 | 75.51 | 74.96 | 76.71 |
| Std. | 0.93 | 0.65 | 0.90 | 0.66 | 0.81 | 0.68 | 0.49 |
| Over. | 4.75 | 4.62 | 4.47 | 4.39 | 4.25 | 3.89 | 2.24 |
| **FMNIST** | | | | | | | |
| Accuracy | 81.79 | 83.40 | 83.64 | 83.70 | 83.85 | 83.75 | 83.87 |
| Std. | 0.06 | 0.06 | 0.12 | 0.04 | 0.11 | 0.11 | 0.05 |
| Over. | 1.89 | 1.85 | 1.78 | 1.75 | 1.70 | 1.56 | 0.88 |

## 6  Discussions

We propose a new direct training algorithm for SNNs that establishes a formal connection between the standard surrogate methods and the zeroth order method applied locally to the neurons. The method introduces systematic randomness in the training that helps in better generalization. The method simultaneously lends itself to efficient back-propagation. We experimentally demonstrate the efficiency of the proposed method in terms of speed-up obtained in training under specialized implementations and its top generalization performance when combined with other training methods, ameliorating their respective strengths.

## Acknowledgement

This work is part of the research project "ENERGY-BASED PROBING FOR SPIKING NEURAL NETWORKS" performed at Mohamed bin Zayed University of Artificial Intelligence (MBZUAI), in collaboration with Technology Innovation Institute (TII) (Contract No. TII/ARRC/2073/2021).

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

# Appendix: Direct Training of SNN using Local Zeroth Order Method

## Appendix A    Proofs of theoretical results

**Lemma 1.** *Assume further that $\int_0^\infty z^{\alpha+1}\lambda(z)dz < \infty$. Then, $\mathbb{E}_{z\sim\lambda}[G^2(u; z, \delta)]$ is a surrogate function.*

*Proof.* Based on our remark above, the only thing left to prove is that the integral $\int_{-\infty}^\infty \mathbb{E}_{z\sim\lambda}[G^2(u; z, \delta)]du$ is finite. To this end, we have (by using equation (8))

$$\int_{-\infty}^\infty \mathbb{E}_{z\sim\lambda}[G^2(u; z, \delta)]du = \int_{-\infty}^\infty \frac{1}{\delta}\int_{\frac{|u|}{\delta}}^\infty z^\alpha\lambda(z)dzdu = \frac{2}{\delta}\int_0^\infty \int_{\frac{|u|}{\delta}}^\infty z^\alpha\lambda(z)dzdu$$

$$= \frac{2}{\delta}\int_0^\infty \int_0^{|z|\delta} |z|^\alpha\lambda(z)dudz = 2\int_0^\infty z^{\alpha+1}\lambda(z)dz,$$

which proves the lemma, as by assumptions the resulting integral is finite.    □

**Theorem 2.** *Let $\lambda$ be a distribution and $\lambda(t)$ its corresponding PDF. Assume that integrals $\int_0^\infty t^\alpha\lambda(t)dt$ and $\int_0^\infty t^{\alpha+1}\lambda(t)dt$ exist and are finite. Let further $\tilde\lambda$ be the distribution with corresponding PDF function*

$$\tilde\lambda(z) = \frac{1}{c}\int_{|z|}^\infty t^\alpha\lambda(t)dt,$$

*where c is the scaling constant (such that $\int_{-\infty}^\infty \tilde\lambda(z)dz = 1$). Then,*

$$\mathbb{E}_{z\sim\lambda}[G^2(u; z, \delta)] = \frac{d}{du}\mathbb{E}_{z\sim\tilde\lambda}[c\,h(u+\delta z)].$$

*Proof.* We have

$$\frac{d}{du}\mathbb{E}_{z\sim\tilde\lambda}[c\,h(u+\delta z)] = c\frac{d}{du}\int_{-\infty}^\infty h(u+\delta z)\tilde\lambda(z)dz = c\frac{d}{du}\int_{-\frac{u}{\delta}}^\infty \tilde\lambda(z)dz = \frac{c}{\delta}\tilde\lambda(-\frac{u}{\delta}) = \frac{c}{\delta}\tilde\lambda(\frac{u}{\delta}),$$

which coincides with (8).    □

For our following result, note that a surrogate function is differentiable almost everywhere, which follows from the Lebesgue theorem on the differentiability of monotone functions. So, taking derivatives here is understood in an "almost everywhere" sense.

**Theorem 3.** *Let $g(u)$ be a surrogate function. Suppose further that $c = -2\delta^2\int_0^\infty \frac{1}{z^\alpha}g'(z\delta)dz < \infty$ and put $\lambda(z) = -\frac{\delta^2}{cz^\alpha}g'(z\delta)$ (so that $\lambda(z)$ is a PDF). Then,*

$$c\,\mathbb{E}_{z\sim\lambda}[G^2(u; z, \delta)] = \mathbb{E}_{z\sim\lambda}[c\,G^2(u; z, \delta)] = g(u).$$

*Proof.* Let us assume that $u \geq 0$ (the other case is similar). Then,

$$\mathbb{E}_{z\sim\lambda}[cG^2(u; z, \delta)] = \frac{c}{\delta}\int_{\frac{u}{\delta}}^\infty z^\alpha\lambda(z)dz = -\frac{1}{\delta}\int_{\frac{u}{\delta}}^\infty z^\alpha\frac{\delta^2}{z^\alpha}g'(z\delta)dz$$

which after change of variables $u = \delta z$ becomes $g(u)$ and finishes our proof.    □

### A.1    Obtaining full-surrogates on Expectation

We demonstrate performance of LOCALZO over different distributions of $z$, such as standard Normal, Uniform($[\sqrt{3}, \sqrt{3}]$) and Laplace($0, \frac{1}{\sqrt{2}}$), for $m \in \{1, 5\}$ and $\delta = 0.05$. The distributions are of unit variance so that parameter $\delta$ is comparable across the methods. We supply SPARSEGRAD algorithm the back-propagation threshold $\tilde B_{th}$ obtained in Table 1. Table 5 shows the performance of the methods on the N-MNIST dataset in terms of accuracy and speedup. The LOCALZO method obtains better train and test accuracies for all cases with a slight compromise in the speedup, except for the uniform distribution where it offers better speedup for $m = 1$ compared to the SPARSEGRAD method.

### A.1.1 From standard Gaussian

Recall that the standard normal distribution $N(0, 1)$ has PDF of the form $\frac{1}{\sqrt{2\pi}} \exp(-\frac{z^2}{2})$. Consequently, it is straightforward to obtain

$$\mathbb{E}_{z \sim \lambda}[G^2(u; z, \delta)] = \frac{1}{\sqrt{2\pi}} \int_{-\infty}^{\infty} \frac{|z|}{2\delta} \exp(-\frac{z^2}{2}) dz = \frac{1}{\delta\sqrt{2\pi}} \exp(-\frac{u^2}{2\delta^2}). \tag{11}$$

### A.1.2 From Uniform Continuous

Consider the PDF of a continuous uniform distribution:

$$f(z; a, b) = \begin{cases} \frac{1}{b-a} & \text{for } z \in [a, b] \\ 0 & \text{otherwise}, \end{cases}$$

where $a < b$ are some real numbers. For the distribution to be even and the resulting scaling constant of the surrogate to be 1 (which translates to $\mathbb{E}[z] = 0$ and $\mathbb{E}[z^2] = 1$, respectively) we set, $a = -\sqrt{3}$, $b = \sqrt{3}$. Then,

$$\mathbb{E}_{z \sim \lambda}[G^2(u; z, \delta)] = \int_{-\infty}^{\infty} \frac{|z|}{2\delta} f(z) dz$$

$$= \frac{1}{2\sqrt{3}} \left[ \int_{-\sqrt{3}}^{-\frac{|u|}{\delta}} \frac{|z|}{2\delta} dz + \int_{\frac{|u|}{\delta}}^{\sqrt{3}} \frac{|z|}{2\delta} dz \right] = \frac{1}{4\sqrt{3}\delta} z^2 \Big|_{\frac{|u|}{\delta}}^{\sqrt{3}}$$

$$= \begin{cases} \frac{1}{4\sqrt{3}\delta}(3 - \frac{u^2}{\delta^2}) & \text{if } \frac{|u|}{\delta} < \sqrt{3}, \\ 0 & \text{otherwise}. \end{cases} \tag{12}$$

### A.1.3 From Laplacian Distribution

The PDF of Laplace distribution is given by:

$$f(z; \mu, b) = \frac{1}{2b} \exp(-\frac{|z - \mu|}{b})$$

with mean $\mu$ and variance $2b^2$. Setting, $b = \frac{1}{\sqrt{2}}$ and $\mu = 0$ and using (5) we obtain,

$$\mathbb{E}_{z \sim \lambda}[G^2(u; z, \delta)] = \frac{2}{\sqrt{2}} \int_{\frac{|u|}{\delta}}^{\infty} \frac{|z|}{2\delta} \exp(-\sqrt{2}|z|) dz = \frac{1}{\delta\sqrt{2}} \int_{\frac{|u|}{\delta}}^{\infty} z \exp(-\sqrt{2}z) dz$$

$$= -\frac{1}{\delta\sqrt{2}} (\frac{z}{\sqrt{2}} + \frac{1}{2}) \exp(-\sqrt{2}z) \Big|_{\frac{|u|}{\delta}}^{\infty} = \frac{1}{2\delta}(\frac{|u|}{\delta} + \frac{1}{\sqrt{2}}) \exp(-\sqrt{2}\frac{|u|}{\delta}). \tag{13}$$

## A.2 Simulating a specific Surrogate

### A.2.1 Sigmoid

Consider the Sigmoid surrogate function, where the Heaviside is approximated by the differentiable Sigmoid function [43]. The corresponding surrogate gradient is given by,

$$\frac{dx}{du} = \frac{d}{du} \frac{1}{1 + \exp(-ku)} = \frac{k \exp(-ku)}{(1 + \exp(-ku))^2} =: g(u)$$

and,

$$g'(u) = -\frac{k^2 \exp(-ku)(1 - \exp(-ku))}{(1 + \exp(-ku))^3}$$

Observe that $g(u)$ satisfies our definition of a surrogate ($g(u)$ being even, non-decreasing on $(-\infty, 0)$ and $\int_{-\infty}^{\infty} g(u) du = 1 < \infty$). Thus, according to Theorem 3, we have

$$c = -2\delta^2 \int_0^{\infty} \frac{g'(t\delta)}{t} dt = 2\delta^2 k^2 \int_0^{\infty} \frac{\exp(-k\delta t)(1 - \exp(-k\delta t))}{t(1 + \exp(-k\delta t))^3} dt = \frac{\delta^2 k^2}{a^2},$$

where, $a := \sqrt{\frac{1}{0.4262}}$. The corresponding PDF is given by

$$\lambda(z) = -\frac{\delta^2}{c}\frac{g'(\delta t)}{z} = a^2\frac{\exp(-k\delta z)(1 - \exp(-k\delta z))}{z(1 + \exp(-k\delta z))^3} \tag{14}$$

Note that the temperature parameter $k$ comes from the surrogate to be simulated, while $\delta$ is used by LOCALZO. We compute the expected back-propagation threshold of SPARSEGRAD for $m = 1$ as, $\tilde{B}_{th} = \delta \, \mathbb{E}_{z \sim \lambda}[|z|]$, with,

$$\mathbb{E}_{z \sim \lambda}[|z|] = 2a^2 \int_0^\infty z\frac{\exp(-az)(1 - \exp(-az))}{z(1 + \exp(-az))^3}dz = \frac{a}{2} = 0.7659. \tag{15}$$

### A.2.2 Fast Sigmoid

Consider also the Fast Sigmoid surrogate gradient [43, 31] that avoids computing the exponential function in Sigmoid to obtain the gradient:

$$\frac{dx}{du} = \frac{1}{(1 + k|u|)^2} =: g(u).$$

We choose $\alpha = -1$ (note that $\alpha = 1$ does not work in this case) and apply theorem 3 so that,

$$c = -2\delta^2 \int_0^\infty \frac{1}{z^\alpha}g'(z\delta)dz = 4\delta^2 \int_0^\infty \frac{1}{z^\alpha}\frac{k\,\mathrm{sign}(z\delta)}{(1 + k|z\delta|)^3}dz$$

$$= 4\delta^2 k \int_0^\infty \frac{z}{(1 + k\delta z)^3}dz = \frac{4}{k} \int_0^\infty \frac{t}{(1 + t)^3}dt = \frac{2}{k}.$$

The PDF is then given by

$$\lambda(z) = -\frac{1}{c}\frac{\delta^2}{z^\alpha}g'(z\delta) = k^2\delta^2\frac{z\,\mathrm{sign}(z\delta)}{(1 + k|z\delta|)^3}. \tag{16}$$

To compute the expected back-propagation threshold, we note,

$$\tilde{B}_{th} = \delta\mathbb{E}_{z \sim \lambda}[|z|] = 2\delta^3 k^2 \int_0^\infty \frac{z^2\,\mathrm{sign}(z\delta)}{(1 + k|z\delta|)^3}dz = 2\delta^3 k^2 \int_0^\infty \frac{z^2}{(1 + kz\delta)^3}dz = \frac{2}{k} \int_0^\infty \frac{x^2}{(1 + x)^3}dx$$

The above integral does not converge. However, if we consider finite support [-a, a], we may compute, $\frac{2}{k}\int_0^a \frac{x^2}{(1+x)^3}dx$

### A.2.3 Inverse Transform Sampling

To simulate a given surrogate in LOCALZO, one needs to sample from the corresponding distribution described by the PDF $\lambda$. Given a sample $r \sim \mathrm{Unif}([0, 1])$ and the inverse CDF $\Lambda^{-1}$ of the distribution, the inverse sampling technique returns, $\Lambda^{-1}(r)$, as a sample from the distribution. Suppose the inverse CDF is not computable analytically from the PDF (or not implementable practically). In that case, we may choose a finite support over which the PDF is evaluated at a sufficiently dense set of points and compute the discretized CDF using the Riemann sum. The inverse discretized CDF is then computed empirically and stored as a list for a finite number of points (spaced regularly) between $[0, 1]$. Sampling from the uniform distribution then amounts to randomly choosing the indices of the list and picking the corresponding inverse CDF values.

### A.3 Expected Back-propagation Thresholds

In what follows, $m$ is the number of samples used in (6), while $k$ is the index of a particular sample. To compute the expected back-propagation threshold, we observe that a neuron is inactive in LOCALZO back-propagation if,

$$|u_i^{(l)}[t] - u_{th}| > |z_k|\delta, \quad \text{for } k = 1, \ldots, m,$$

$$\text{or,}|u_i^{(l)}[t] - u_{th}| > t\delta, \text{ where } t = \max_k\{|z_1|, \cdots, |z_m|\}$$

Assume $z_k \sim \lambda$, where $\lambda(t)$ denotes the PDF of the sampling distribution, with the corresponding CDF denoted by $F_{z_k}$. The PDF, $\tilde{\lambda}$, of the random variable $|z_k|$ is given by

$$\tilde{\lambda}(x) = \begin{cases} 0, & \text{if } x < 0 \\ 2\lambda(x), & \text{otherwise.} \end{cases} \tag{17}$$

The corresponding CDF is obtained by integrating the previous expression,

$$F_{|z_k|}(x) = \begin{cases} 0, & \text{if } x < 0 \\ 2(F_{z_k}(x) - F_{z_k}(0)), & \text{otherwise.} \end{cases} \tag{18}$$

Further note that,

$$F_t(x) = P(t < x) = \prod_{k=1}^{m} P(|z_k| < x) = F_{|z_k|}^m(x) \tag{19}$$

If we denote the PDF of the random variable $t$ as $\hat{\lambda}$, we obtain

$$\hat{\lambda}(x) = m F_{|z_k|}^{m-1}(x)\tilde{\lambda}(x). \tag{20}$$

Finally, the expected back-propagation threshold takes the form

$$\tilde{B}_{th} = \delta \mathbb{E}[t] = \delta \int_0^\infty t\hat{\lambda}(t)dt. \tag{21}$$

In the cases of distributions used in experimental sections, the previous expression simplifies. Table 1 gives the numerical values for some particular $m$. To obtain an expected back-propagation threshold, we would like to evaluate:

$$\tilde{B}_{th} = \delta \mathbb{E}[t] = \delta \int_0^\infty t\hat{\lambda}(t)dt = \delta m \int_0^\infty t F_{|z|}^{m-1}(t)\tilde{\lambda}(t)dt$$

For the standard normal distribution, $\lambda = \text{Normal}(0, 1)$ we have, $F_{|z|}(t) = \text{erf}(\frac{t}{\sqrt{2}})$ giving,

$$\tilde{B}_{th} = \frac{2\delta m}{\sqrt{2\pi}} \int_0^\infty t\,\text{erf}^{m-1}(\frac{t}{\sqrt{2}})\exp(-\frac{t^2}{2})dt. \tag{22}$$

For uniform continuous, $\lambda = \text{Unif}([-\sqrt{3}, \sqrt{3}])$, we have, $F_{|z|}(t) = \frac{t}{\sqrt{3}}$ giving,

$$\tilde{B}_{th} = \frac{\delta m}{\sqrt{3}} \int_0^{\sqrt{3}} t(\frac{t}{\sqrt{3}})^{m-1}dt = \delta\sqrt{3}\frac{m}{m+1}. \tag{23}$$

For Laplace distribution, $\lambda = \text{Laplace}(0, \frac{1}{\sqrt{2}})$, we have, $F_{|z|}(t) = 1 - \exp(-\sqrt{2}t)$,

$$\tilde{B}_{th} = \delta m\sqrt{2} \int_0^\infty t(1 - \exp(-\sqrt{2}t))^{m-1}\exp(-\sqrt{2}t)dt. \tag{24}$$

## Appendix B  Additional Results

### B.1  Computational Speedup in SPARSEGRAD

The SPARSEGRAD frame-work derives speedup by performing the back-propagation in a layerwise fashion. To summarize their finding, let us start with eqn. 1.

$$u_i^{(l)}[t] = \beta u_i^{(l)}[t-1] + \sum_j w_{ij}x_j^{(l-1)}[t] - x_i^{(l)}[t-1]u_{th}$$

which, after unfolding the recurrence and using the fact that $u_i^{(l)}[0] = 0$, can be restated as:

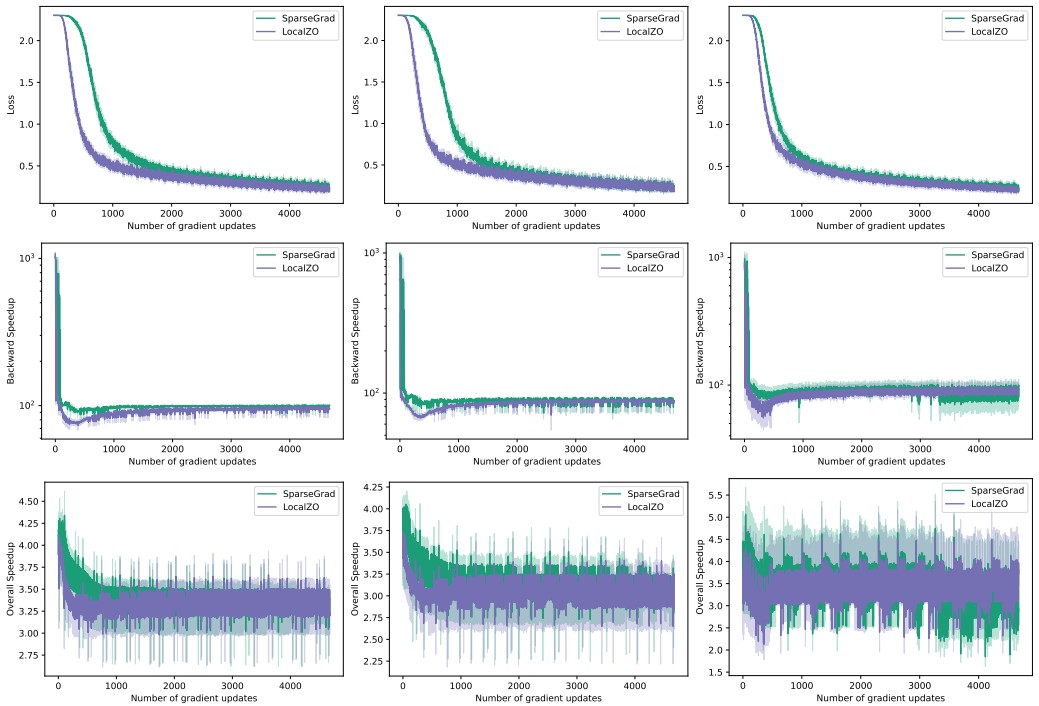

Figure 4: We plot training loss, backward speedup (Back.), and overall speedup (Over.) of SPARSEG-RAD and LOCALZO after each gradient update performed on NMNIST data, as summarized in Table 5 . The columns represent plots for three distributions Normal$(0, 1)$, Laplace$(1, \frac{1}{\sqrt{2}})$, and Unif$([-\sqrt{3}, \sqrt{3}])$ respectively, with $\delta = 0.05$ and $m = 1$. The LOCALZO algorithm converges faster than the SPARSEGRAD method and provides comparable backward and overall speedup.

$$u_i^{(l)}[t] = \sum_j \sum_{k=0}^{t-1} \beta^{t-k-1} x_j^{(l-1)}[k] w_{ij} - \sum_{k=0}^{t-1} \beta^{t-k-1} x_i^{(l)}[k] u_{th} \tag{25}$$

where, the **input trace**, $\sum_{k=0}^{t-1} \beta^{t-k-1} x_j^{(l-1)}[k]$ can be computed in a layer-wise fashion from the forward propagation. Note that, by ignoring the reset mechanism in the gradient the computation, the gradient of the loss can be written as:

$$\frac{\partial l}{\partial w_{ij}} = \sum_t \frac{\partial l[t]}{\partial x[t]} \frac{\partial x[t]}{\partial u[t]} \frac{\partial u[t]}{\partial w_{ij}}$$

$$= \sum_t \frac{\partial l[t]}{\partial x[t]} \frac{\partial x[t]}{\partial u[t]} \sum_{k=0}^{t-1} \beta^{t-k-1} x_j^{(l-1)}[k] \tag{26}$$

where, the term $\frac{\partial l[t]}{\partial x[t]}$ is coming from the next layer, the term $\frac{\partial x[t]}{\partial u[t]}$ is the given by 5, so it can be computed in the forward propagation along with the **input trace**. Thus, whenever, many neurons are inactive, i.e., $\frac{\partial x[t]}{\partial u[t]} = 0$ the SPARSEGRAD frame-work can reduce the computation burden of the back-propagation.

## B.2 Further comparison with SPARSEGRAD

In the section 4.4, we derived surrogates corresponding to distributions and distributions corresponding to popular surrogates. We implement LOCALZO, with $\delta = 0.05$ for different distribution

Table 5: Performance comparison on NMNIST for m=1

| METHOD | TRAIN | TEST | BACK. | OVER. |
|---|---|---|---|---|
| $z \sim$ NORMAL$(0,1), \delta = 0.05, m = 1$ | | | | |
| SPARSEGRAD | $93.26 \pm 0.31$ | $91.86 \pm 0.29$ | 99.57 | 3.38 |
| LOCALZO | $94.38 \pm 0.12$ | $93.29 \pm 0.08$ | 92.27 | 3.34 |
| $z \sim$ LAPLACE$(1, \frac{1}{\sqrt{2}}), \delta = 0.05, m = 1$ | | | | |
| SPARSEGRAD | $93.97 \pm 0.43$ | $92.65 \pm 0.52$ | 88.2 | 3.19 |
| LOCALZO | $94.25 \pm 0.17$ | $93.05 \pm 0.09$ | 83.7 | 3.07 |
| $z \sim$ UNIF$([-\sqrt{3}, \sqrt{3}]), \delta = 0.05, m = 1$ | | | | |
| SPARSEGRAD | $93.34 \pm 0.44$ | $91.85 \pm 0.35$ | 83.2 | 3.26 |
| LOCALZO | $94.24 \pm 0.46$ | $93.05 \pm 0.37$ | 84.8 | 3.43 |
| SIGMOID, $\delta = 0.05, k \approx 30.63, m = 1$ | | | | |
| SPARSEGRAD | $92.96 \pm 0.26$ | $91.04 \pm 0.32$ | 87.45 | 3.00 |
| LOCALZO | $93.98 \pm 0.08$ | $92.97 \pm 0.05$ | 83.54 | 3.02 |
| FASTSIGMOID, $\delta = 0.05, k = 100, m = 1$ | | | | |
| SPARSEGRAD | $93.24 \pm 0.23$ | $92.16 \pm 0.20$ | 84.87 | 3.18 |
| LOCALZO | $93.44 \pm 0.13$ | $92.52 \pm 0.09$ | 73.23 | 3.11 |

Table 6: Performance comparison on NMNIST for m=5

| METHOD | TRAIN | TEST | BACK. | OVER. |
|---|---|---|---|---|
| $z \sim$ NORMAL$(0,1), \delta = 0.05, m = 5$ | | | | |
| SPARSEGRAD | $95.02 \pm 0.29$ | $93.39 \pm 0.25$ | 80.0 | 3.40 |
| LOCALZO | $95.20 \pm 0.22$ | $93.69 \pm 0.17$ | 77.7 | 3.22 |
| $z \sim$ LAPLACE$(1, \frac{1}{\sqrt{2}}), \delta = 0.05, m = 5$ | | | | |
| SPARSEGRAD | $94.73 \pm 0.29$ | $93.13 \pm 0.23$ | 72.9 | 3.15 |
| LOCALZO | $95.07 \pm 0.03$ | $93.63 \pm 0.05$ | 69.4 | 2.80 |
| $z \sim$ UNIF$([-\sqrt{3}, \sqrt{3}]), \delta = 0.05, m = 5$ | | | | |
| SPARSEGRAD | $94.82 \pm 0.27$ | $93.38 \pm 0.17$ | 76.4 | 3.14 |
| LOCALZO | $94.95 \pm 0.35$ | $93.47 \pm 0.23$ | 73.5 | 2.91 |

such as Normal$(0,1)$, Laplace$(1, \frac{1}{\sqrt{2}})$, and Unif$([-\sqrt{3}, \sqrt{3}])$. They all have a unit variance to ensure $\delta$ is comparable across the distributions. The corresponding back-propagation thresholds for SPARSEGRAD are derived in Table 1.

For the Sigmoid surrogate, we take the temperature parameter $k = a/\delta \approx 30.63$ so that $c = \frac{\delta^2 k^2}{a^2} = 1$ for the corresponding PDF derived in eqn.14. We supply SPARSEGRAD method the corresponding back-propagation threshold, $\tilde{B}_{th} = 0.766\delta$, as obtained in eqn. 15.

For the Fast Sigmoid surrogate, we choose $k = 100$ following [31] so that $c = \frac{2}{k}$. To compute the expected back-propagation threshold, we consider finite support $[-10, 10]$ used in the inverse transform sampling of $z$ and evaluate $\tilde{B}_{th} = 0.0461$. Table 5 reports accuracies obtained by LOCALZO and SPARSEGRAD across various sampling distributions, with $m = 1$. LOCALZO offers better accuracies compared to SPARSEGRAD across all the distributions, with a slight compromise in speed-up in most cases. For uniform distribution, LOCALZO offers even better speed-up than SPARSEGRAD.

Table 6 reports the details of the comparison over the N-MNIST dataset for sampling $z$ from various distributions with $m = 5$. LOCALZO method obtains better test accuracies. The SPARSEGRAD method is supplied with the respective surrogate and back-propagation threshold as computed in

Table 7: Hyper-parameter settings for general comparison

|  | CIFAR-10/100 | ImageNet-100 | DVS-CIFAR-10 | DVS-Gesture | N-Caltech/NCARS |
|---|---|---|---|---|---|
| Number epochs | 300 | 300 | 300 | 200 | 200 |
| Mini batch size | 64 | 64, 72 | 64 | 64 | 16 |
| T | 6,4,2 | 4 | 10 | 10 | 10 |
| LIF: $\beta$ | 0.5 | 1 | 0.5 | 0.5 | 0.5 |
| LIF: $u_0$ | 0 | 0 | 0 | 0 | 0 |
| LIF: $u_{th}$ | 1 | 1 | 1 | 1 | 1 |
| LOCALZO: $\delta$ | 0.5 | 0.5 | 0.5 | 0.5 | 0.5 |
| LOCALZO: m | 5 | 5, 20 | 1 | 5 | 5 |
| LOCALZO: $\lambda$ | N(0,1) | N(0,1) | N(0,1) | N(0,1) | N(0,1) |
| $\lambda_{TET}$ | 0.05 | 0.001 | 0.0001 | 0.05 | 0.05 |
| Learning Rate | 0.001 | 0.1 | 0.001 | 0.001 | 0.001 |

Optimizer: Adam with betas: (0.9; 0.999), Rate Scheduler: cosine annealing

Table 8: Hyper-parameter settings for comparison in SPARSEGRAD framework

|  | FMNIST | SHD | N-MNIST |
|---|---|---|---|
| Number of Input Neurons | 784 | 1156 | 700 |
| Number of Hidden | 200 | 200 | 200 |
| Number of classes | 10 | 10 | 20 |
| Number epochs | 20 | 20 | 20 |
| Mini batch size | 256 | 256 | 256 |
| T | 100 | 300 | 500 |
| $\Delta t$ | 1ms | 1ms | 2ms |
| $\tau_{\text{eff}}$ | 20ms | N/A | N/A |
| $\theta$ | 0.2 | N/A | N/A |
| $u_0$ | 0 | 0 | 0 |
| $u_{th}$ | 1 | 1 | 1 |
| LOCALZO: $\delta$ | 0.05 | 0.05 | 0.05 |
| Optimiser | Adam | Adam | Adam |
| Learning Rate | 0.0002 | 0.0002 | 0.001 |
| Betas | (0.9; 0.999) | (0.9; 0.999) | (0.9; 0.999) |

Tab.1. The LOCALZO method achieves better accuracies than SPARSEGRAD for all the distributions, though the speed-up is reduced due to higher sampling cost at $m = 5$.

In table 8, we further provide the hyper-parameters for the comparison in the SPARSEGRAD framework, which are replicated from their work. The FMNIST data uses latency encoding, where each input pixel $x$, is converted to a single spike, the spike timing is determined by:

$$T(x) = \left\{ \tau_{\text{eff}} \log \frac{x}{x-\theta} \right.$$

Table 7 gives the hyper-parameter setting for comparison in general framework reported in Tab. 2.

## B.3 Dropout effect in LOCALZO

A fixed neuron has an active gradient for $m = 1$, only if $|u_i[t] - u_{th}| < |z|\delta$ as described in eqn5. Observe that $u_i[t]$ is specific to an input data point and time-step. The same neuron can be inactive for another data-point, or at a different time-step. Moreover, a neuron can have a non-zero gradient with respect to a single data point, even if the spikes are zero. Fixing a neuron and time-step, we plot the distribution of membrane potential (minus membrane threshold), spikes, and non-zero gradients over different data points across batches, where we implement LOCALZO on MNIST data with $\delta = 0.5$. The plot captures the sparsity of spikes and sparsity of neuron gradients for LOCALZO.

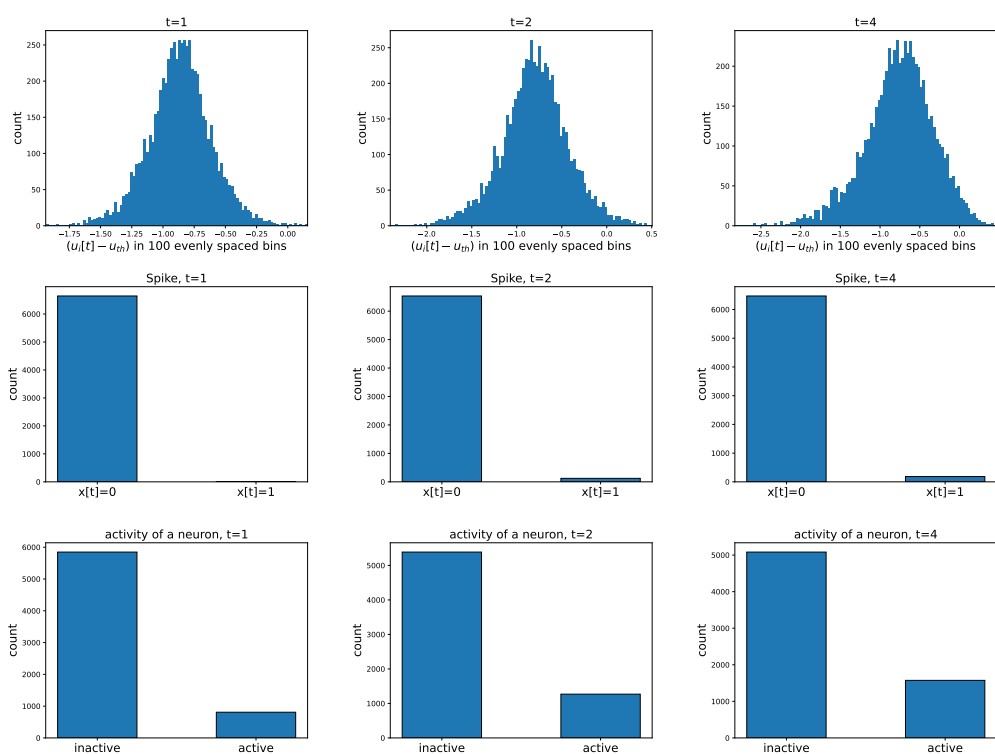

Figure 5: Fixing a neuron, we plot the distribution of membrane potential, spikes, and gradient activity over different data points across batches, where we implement LOCALZO on MNIST data with $\delta = 0.5$. The plot captures the sparsity of spikes and sparsity of neuron gradients for LOCALZO at different time-steps.

