# OpenReview forum: "Direct Training of SNN using Local Zeroth Order Method"
_NeurIPS.cc/2023/Conference — NeurIPS 2023 poster_

### Official Review · Reviewer_Rm6P · 2023-07-02

**Soundness:** 4 excellent
**Presentation:** 3 good
**Contribution:** 3 good
**Rating:** 8
**Confidence:** 3

**Summary:**

This paper proposes a Local Zeroth Order method, which can fit arbitrary surrogate functions by sampling a group of variables from a certain distribution. Experiments have verified the superiority of the proposed scheme.

**Strengths:**

1. The authors‘ idea about fitting arbitrary surrogate functions by sampling is very novel.
2. Theoretical analysis about ZO function is persuasive and profound.
3. Relevant experiments, especially on ResNet-19, can show the obvious advantages of ZO function.

**Weaknesses:**

1. If time permits, I suggest that the authors can supplement their experiments on large-scale datasets (e.g. ImageNet).

**Questions:**

1. I note that the firing threshold used by the authors in the code seems to be a fixed value of 1, so has the expected threshold proposed by the authors in Section 4.5 been used in actual experiments?

**Limitations:**

I think that the proposed ZO function may require more extensive experiments, especially on large-scale datatsets. Overall, I think this paper have met the acceptance standard.

---

> ### Author Rebuttal · Authors · 2023-08-09
>
> **W:** If time permits, I suggest that the authors can supplement their experiments on large-scale datasets (e.g. ImageNet).
>
> **A:** Due to the time constraint, we were not able to run our experiments on the full ImageNet dataset, but we could run them on the Imagenet-100 dataset, which has the same dimensionality as Imagenet. On such dataset, we confirmed that LocalZO is comparable in test accuracy to the standard surrogate. Please refer to our answer to reviewer cVHq for full details on this experiment (such details will also be added in the final version of our paper).
>
> **Q:** I note that the firing threshold used by the authors in the code seems to be a fixed value of 1, so has the expected threshold proposed by the authors in Section 4.5 been used in actual experiments?
>
> **A:** Actually, throughout the paper we ambiguously use the same term ``threshold'' for two different values. One is the firing threshold ($u_{th}$) for the membrane potential in order for a neuron to spike (which in the code is set to 1), the other has been the threshold for the backpropagation ($\tilde{B}_{th}$) of gradients, to which the results of Section 4.5 pertain. These thresholds were used in the experiments where we compare our method with SparseGrad method.

---

> > ### Comment · Reviewer_Rm6P · 2023-08-11
> > **Response to rebuttal**
> >
> > I would like to thank the authors to address all my concerns, especially regarding the clarification on the term threshold used in the paper, now this is clear to me. For the experiments, due to the time limit, I am satisfied that the authors provide more results on more datasets like the Imagenet-100 dataset.
> >
> > In summary, I am happy to the response and would like to increase my rating from 7 to 8.

---

### Official Review · Reviewer_8Gkq · 2023-07-05

**Soundness:** 4 excellent
**Presentation:** 3 good
**Contribution:** 3 good
**Rating:** 7
**Confidence:** 3

**Summary:**

The authors propose a new method for training spiking neural networks
(SNNs). To estimate the gradient of the step function for spike
generation, they propose to directly estimate the gradient by local
sampling around the point of derivation and averaging the linearly
calculated slopes, a method that is known in other fields as
zeroth-order estimation of the gradient of an arbitrary (potentially
non continuous) function. They show that, if the distribution used for
sampling is chosen appropriately, the approximated gradient approaches
in expectation the common surrogate gradient approach, where an
arbitrary surrogate function is used for smoothening the gradient of
the step function.  They further show how to choose the sampling
distribution for a couple of given surrogate functions. Finally, they
show that their approach outperforms previous methods on common
benchmark tasks in terms of accuracy. In particular, gradient
calculation is sparser (since gradients are deemed zero for larger
sampled values) in particular if only few values are sampled, and thus
more efficiently to compute (unless one truncates the surrogate, which
performs similarly in this regard).


**Strengths:**

The paper makes a very interesting connection of the surrogates used
to train SNNs to a sampling based zeroth order method, and shows that
and how (in theoretical depth) samples can be generated to approximate
any surrogate used in literature.

While the method does not significantly changes the state-of-the-art,
since the SparseGrad method (that truncates the surrogates to have a
sparser backward pass) seem to perform very similar in accuracy
as well as compute efficiency, it nevertheless provides a very
interesting theoretical underpinning of how sparseness can be
introduced into the backward pass.

The paper is also very well written, clearly motivated and
provides clear mathematical proofs of their claims.


**Weaknesses:**

While the theoretical connection is interesting it could be argued
that the existing SparseGrad method performs similarly in practice, so that there is not much of an improvement of the state-of-the-art. However, still, the accuracy seems to be slightly improved due to the
introduced randomness that apparently leads to slightly better
generalization.



**Questions:**

The general approach is motivated by dropout, where activity is
randomly changed in the forward pass. However, the authors actually
"apply" this principle only for the backward pass (as demanded by
the zeroth-order approach). Since $m=1$ for most cases in the
experiments, I wonder how results would change if spikes in the
forward pass would be also randomly generated (and not only the
(pseudo) backward spikes), in other words, when the pseudo-backward
spikes where actually also the forward spikes. This would be similar
to neuron with random spike generation, which actually is biologically
very realistic. The randomness in the forward pass would effectively
smooth over the Heaviside function as well. However, maybe in practice
the activity profile of the SNN would be impacted too much. That might
be an interesting discussion point.


**Limitations:**

It would have been interesting to discuss / compare the methods with SNNs that use some form of random spike generation.

---

> ### Author Rebuttal · Authors · 2023-08-09
>
>  **W:** While the theoretical connection is interesting it could be argued that the existing SparseGrad method performs similarly in practice, so that there is not much of an improvement of the state-of-the-art. However, still, the accuracy seems to be slightly improved due to the introduced randomness that apparently leads to slightly better generalization.
>
> **A:** As suggested by one of the reviewers, we further tested our method on several neuromorphic datasets that we did not consider previously. We hope that these new results emphasize even further the advantage of LocalZO method compared to surrogate gradient. Please refer to our reply to the reviewer cVHq for the results and experimental setting.
>
> **Q:** The general approach is motivated by dropout, where activity is randomly changed in the forward pass. However, the authors actually "apply" this principle only for the backward pass (as demanded by the zeroth-order approach). Since for most cases in the experiments, I wonder how results would change if spikes in the forward pass would be also randomly generated (and not only the (pseudo) backward spikes), in other words, when the pseudo-backward spikes where actually also the forward spikes. This would be similar to neuron with random spike generation, which actually is biologically very realistic. The randomness in the forward pass would effectively smooth over the Heaviside function as well. However, maybe in practice the activity profile of the SNN would be impacted too much. That might be an interesting discussion point.
>
> **A:** This is a rather curious point and one that we considered and plan to consider. As you pointed out, a neuron which is a random spike generator is biologically plausible concept, but also, on the ANN side of the story, there are results which show that ReLU or Leaky ReLU with slopes sampled from some distribution during training, while during inference one takes the expected slope, show improved performance over their respective rigid versions.
>
> To introduce randomness in the forward pass along the above lines, one can simply, at each time step, randomly sample threshold, i.e. $u_{th}\sim \lambda$, where $\lambda$ is some distribution ideally with finite support contained in positive real numbers.  Already here, there is a choice for what is the output of the neuron. For example, first natural choice is $H(u[t]-u_{th})$ ($H$ being the Heaviside function), while in the backward pass one uses the any surrogate applied to the input $|u[t]-u_{th}|$. First tests of this choice (VGG16, CIFAR10 dataset (no data augmentation), 60 epochs) show that the performance is comparable to that of the plain setting, but we noticed that the train and test accuracy are kept close to each other (while in the plain setting the network tends to overfit early on), so one may suspect that even in this simple setting the model is able to generalize well whatever it learned (of course, this is a rather simple experiment and one should not take any results or conclusions as definitive).
> A more interesting, in our modest opinion, is the situation where the neuron outputs $u_{th}*H(u[t]-u_{th})$. As you put it, the randomness in the forward pass would effectively smooth over the Heaviside function as well, seems to be satisfied in this situation. Moreover, all sorts of new phenomena arise. One can study the probability of a neuron (having fixed membrane potential) to fire, the expected output of a neuron, distribution of outputs. At the same time, in the backward pass one faces some interesting choices for the surrogate gradients. If we follow the settings of the previous experiment and use some fixed surrogate, it seems that the network is learning slightly worse than in the other setting (with similar generalization property). But, it also seems that there should be a natural choice for the surrogate, which, intuitively speaking, should depend on the two dimensional distribution of ``neuron membrane potential - neuron output'' (the situation should however be compared with the setting of probabilistic spiking neural networks, c.f. https://arxiv.org/pdf/1910.01059.pdf).
>
> In conclusion, we firmly agree with your suggestion of introducing the randomness in the forward pass, and moreover to introduce it in such a way that the backward gradient becomes ``naturally apparent'' rather than becoming a choice. The potential elegance of this situation requires a detailed understanding and extensive experiments, that we hope to pursuit in the future.

---

> > ### Comment · Reviewer_8Gkq · 2023-08-12
> >
> > I thank the authors for the detailed responses.

---

### Official Review · Reviewer_DTqo · 2023-07-07

**Soundness:** 3 good
**Presentation:** 3 good
**Contribution:** 2 fair
**Rating:** 4
**Confidence:** 4

**Summary:**

This paper proposes a new direct training algorithm for SNN, combining the standard surrogate methods and zeroth order method together. The algorithm applies the 2-point zeroth order method on the Heaviside function to generate a surrogate gradient, which is more efficient. The author applied his method to various dataset such as CIFAR-10 and CIFAR-100 and outperforms the SOTA methods.

**Strengths:**

1.	Due to the efficiency of zeroth order method, this algorithm is computational friendly, reducing the computational burden significantly. The gradients are only back propagated through active neurons.
2.	According to theorem 3, the algorithm is able to simulate arbitrary surrogate, which means this algorithm has great expression capability.

**Weaknesses:**

1.This paper has limited novelty. It seems simply to be a combination the forward gradient method [1] and sparse gradient together [2].

2.Worry about the scalability of the proposed method, as forward gradient method is not widely applicable to large networks [1].

3.The experiments only show marginal improvements.




[1] SCALING FORWARD GRADIENT WITH LOCAL LOSSES. ICLR 2023
[2] Sparse Spiking Gradient Descent. NeurIPS 2021.

**Questions:**

How to choose the distribution function of z for different tasks?

**Limitations:**

No. The author should discuss the limitation of the scalability of the forward gradient method in large network.

---

> ### Author Rebuttal · Authors · 2023-08-09
>
> **W1:** This paper has limited novelty. It seems simply to be a combination the forward gradient method [1] and sparse gradient together [2].
>
> **A1:** We try to elaborate on the motivation and technical implementation of our work, hoping to be more clear why our method is not simply a combination of two methods you mentioned.
>
> 1) Loosely speaking, forward gradient method is used to estimate the updates of the weights of the network by observing the change in the outputs of the network with respect to the perturbation of the weights. In a variant of this general principle, the authors in [1] use somewhat local version, where the gradients of the weights are estimated by the perturbation of the activations of the subsequent postsynaptic neurons and learning is performed in combination with local losses.
>
>     Contrary to these principles, zeroth-order in our method stems from a different motivation and is focused solely on the spiking neuron's activation function (i.e. the Heaviside function) and its gradient. In general, one uses a fixed surrogate gradient, i.e. a function which will serve in the backward pass in place of the non-existent (in the classical sense) derivative of the Heaviside function. For us, the surrogate function changes from time step to time step, and from neuron to neuron, depending on the ``activity'' of the neuron itself (please refer to the motivation section of our paper for more details and interpretation of the activity of the neuron). Also, in our work the training is done fully end to end using a loss on the output.
>
> 2) Sparse gradient method [2] proposes to use the surrogate gradients with compact support, demonstrating its advantages in the energy efficiency, as in general there will be abundance of zero gradients that will not be passed in the backward pass. However, once the surrogate function is chosen, it is fixed throughout the training, and although comparable to using surrogate gradients which have non-compact support, it is slightly lagging behind when it comes to performance.
>
>       The surrogate gradients that are present in our method have compact support at each time step, potentially offering the benefits of sparsegrad method. However, we do not fix upfront this surrogate function, but rather it changes from time step to time step, in a somewhat random but controllable way: Random, because it depends on a random sample from a distribution and controllable because the distribution is fixed.
>
>
>      The method we propose is more than sum of its basic parts. Motivated by the effect of randomness-based regularizer such as dropout, our goal was to introduce a direct training method which will have regularizing effects due to some introduced randomness, which will offer the benefits of the sparsegrad method, but at the same time having the ability to simulate both non-compactly and compactly supported surrogate functions, hence keeping the best from both worlds (performance and potential energy efficiency).
>
> 3) Next, we provide both theoretical and practical framework for our method, further establishing its soundness and validity. Our Theorems 2. and 3. provide close connection between distributions that are intrinsic to our method and surrogate functions that are obtained in the expectations showing that LocalZO can be used as a substitute for any surrogate function that is used in SNN literature. You may refer to Section 4.4 for the applications of these results for some of the more commonly used surrogates.
>
> **W2:** Worry about the scalability of the proposed method, as forward gradient method is not widely applicable to large networks [1].
>
> **A2:** As we only apply forward gradient locally, at neuronal level to provide the derivative of the Heaviside function, there is no problem in applying the method to deeper or larger networks. For example, we trained on Imagenet-100 and confirmed that LocalZO is comparable in test accuracy to the standard surrogate. Please refer to our answer to reviewer cVHq for full details on this experiment (such details will also be added in the final version of our paper).
>
> **W3:**  The experiments only show marginal improvements.
>
> **A3:** We tested LocalZO method, as suggested by one of the referees on several other neuromorphic datasets. The method demonstrates its advantage in generalization performance compared to the standard surrogate training. Please refer to our reply to the reviewer cVHq, where we present newly obtained results and the experimental setting.
>
> **Q:** How to choose the distribution function of z for different tasks?
>
> **A:** This is an interesting and, if we may say so, a difficult question. To the best of our knowledge, there is no systematic study of performances of surrogate functions on different tasks, and in our experience in the literature one can find all sort of (reasonable) functions yielding trained models that perform well on various task and datasets (and in fact, different surrogate functions performing well on the same tasks). On the other side, the reviewer may take a look at the paper by Y. Li et al. "Differentiable Spike: Rethinking Gradient-Descent for Training Spiking Neural Networks" where a somewhat related question has been addressed.
>
> [1] SCALING FORWARD GRADIENT WITH LOCAL LOSSES. ICLR'23
>
> [2] Sparse Spiking Gradient Descent. NeurIPS'21.

---

> > ### Comment · Reviewer_DTqo · 2023-08-13
> > **still have some conerns**
> >
> > I appreciate the authors' feedback. I still have some concerns about the paper.
> >
> > 1. Efficiency vs generalization. The proposed method introduces randomness to improve generalization. However, it's highly related to the number of samples of z, while large number of samples may impair computational efficiency. Could authors show a study of different number of samples used in the algorithm for both accuracy and computational cost, so that we can see the separate effect of local forward method and randomness?
> >
> > 2. Scalability of the method. As shown in the rebuttal to reviewer cVHq, the method only gains marginal improvement or even worse result on Imagenet-100 dataset.
> >
> > 3. Lack details of the adaptivity of the zero-order method. I appreciate the theoretical analysis that the proposed method can approximate arbitrary surrogate functions. However, it is not clear to me that the algorithm how to adaptively select the distribution z during training, especially there is no closed form for the expected surrogate function.

---

> > > ### Author Response · Authors · 2023-08-18
> > >
> > > **Q1** Efficiency vs generalization.
> > >
> > > **A1** We provide additional details of the LocalZO method from Table 3 of the paper. The table below shows the test accuracy of the LocalZO method, and overall speedup compared to the surrogate for $m \in {1,3,5,7,10, 20, 100}$, with z sampled from Gaussian distribution, averaged over 5 experiments. We also report the accuracy of the corresponding Gaussian surrogate.
> > >
> > > In general, by increasing m, the method approximates the surrogate better, still offers the regularizing effect and potentially improves the generalization, but also requires more computation. Larger m also leads to more non-zero gradients at the neuronal level in the backward pass, which leads to reduced overall speedup. On the other hand, smaller m introduces higher randomness (less “controlled”) still yielding regularization, which helps obtain better generalization, but as well as the potential speedup.
> > >
> > > In conclusion, m should be treated as a hyper-parameter, its value depending on the training setting itself. In our experiments, we chose m = 1 or 5 for most of the experiments, as a proof of concept, but also because it offers a nice balance between the speed-up as well as performance.  However, here is a more complete table where one can see the effect of m, both in terms of speed-up and accuracy.
> > >
> > > NMNIST , Surrogate Acc: 93.70, std: 0.17
> > > |m|1|3|5|7|10|20|100|
> > > | ---- | ---- | ---- | ---- | ---- | ---- | --- | --- |
> > > | Acc: |  93.29 | 93.61 | 93.69 | 93.66 |  93.76 | 93.67 | 93.81 |
> > > |Std: | 0.08  |  0.15  |   0.17 |   0.13  |   0.14  | 0.08 | 0.14 |
> > > | Speedup |3.33|3.28|3.22|3.16| 3.06 | 2.82 | 1.59 |
> > >
> > > SHD, Surrogate Acc: 75.47, std: 0.69
> > >
> > > |m|1|3|5|7|10|20|100|
> > > | ---- | ---- | ---- | ---- | ---- | ---- | --- | --- |
> > > | Acc | 76.55 | 76.55 | 76.50 | 75.49 | 75.51 |  74.96 |  76.71 |
> > > | std. |  0.93  | 0.65  |  0.90  |  0.66   |  0.81   |  0.68 |  0.49 |
> > > | Speedup | 4.75 | 4.62 |  4.47 |  4.39  | 4.25 |  3.89 | 2.24 |
> > >
> > > FMNIST,  Surrogate Acc: 83.35, std: 0.16
> > >
> > > |m|1|3|5|7|10|20|100|
> > > | ---- | ---- | ---- | ---- | ---- | ---- | --- | --- |
> > > | Acc |  81.79 | 83.40 | 83.64 | 83.70 |  83.85 |  83.75 |  83.87 |
> > > | std. | 0.06   | 0.06 | 0.12 | 0.04 | 0.11 | 0.11 | 0.05 |
> > > | Speedup  | 1.89 | 1.85 | 1.78 | 1.75 |  1.70 | 1.56  | 0.88 |
> > >
> > > Additionally, to evaluate quantitatively the quality of the LocalZO estimator improves with the number of random samples $m$, we provide below a plot of the mean and standard deviation of the distribution of such an estimator, for several values of $m=1, 5, 10, 20$, where the LocalZO estimator is sampled $10^5$ times.
> > >
> > > More precisely, we consider the Gaussian ZO estimator with $\delta=0.5$, and for each coordinate $u$ of the input to the Heaviside (from a grid of 120 values from -3 to 3),  we compute $10^5$ samples $g_i(u)$, $i=\{1, …, 10^5\}$ of the localZO estimator of the gradient as:
> > > $g_i(u) = \\frac{1}{m} \sum_{j=1}^m G^2(z_j)$, with $G^2(z) = \begin{cases}
> > >     \\frac{|z|}{2 \\delta} ~\\text{if} ~|u| \\leq |z| \\delta,
> > >      0 ~ \text{otherwise}
> > > \end{cases} $ (from eqn. [5, 6]), and $z_1, …, z_m$ are i.i.d. samples from a standard normal distribution. We then report the mean (as the main curve) and standard deviation (as the shaded area) of the samples $\{g_1(u), … , g_{10^5}(u)\}$, for each value of $u$.
> > >
> > > As we can observe, after already m=5 samples the standard deviation becomes reasonably low, and it decreases as we progressively increase $m$.
> > >
> > > https://anonymous.4open.science/r/rebtalneurips-D01C/
> > >
> > > **Q2** Scalability of the method.
> > >
> > > **A2:**  It is known that using regularizers such as Dropout require longer training or larger models for their advantages to take effect (see for example references [1, 2]]). As LocalZO method incorporates randomness as a regularizer, the time it takes to show advantages varies on how large the dataset is, or how big the network is. For smaller scale datasets, we have shown in our experiments that the advantages over surrogate are visible after a lower number of epochs. However, for larger datasets, such as Imagenet-100, we need to train for longer epochs to show advantage.
> > > Having in mind our discussion on usage of m in our answer to your first question, we provide the results after training LocalZO on Imagenet-100 dataset as well as surrogate, for 300 epochs, with batch size 72.
> > >
> > > | | Surrogate | LocalZO |
> > > | ---- | -----: | -----: |
> > > | ImageNet-100  | 81.23 | 83.33 |
> > >
> > > As we can see, this time, the advantage of our method is much more emphasized compared to the results previously reported.  We use ImageNet policy with standard augmentation, and m=20 for LocalZO.
> > >
> > > [1] Goodfellow, I., Bengio, Y., and Courville, A. Deep learning. MIT press, 2016.
> > >
> > > [2] Hernández-García, A., & König, P. (2018). Data augmentation instead of explicit regularization

---

> > > > ### Author Response · Authors · 2023-08-18
> > > >
> > > > **Q3** Lack details of the adaptivity of the zero-order method.
> > > >
> > > > **A3**: In our setting, either the distribution to sample z,  or the surrogate we want to simulate with LocalZO is given in advance, and once given, it is fixed throughout the training. So, we do not consider the adaptivity of LocalZO as the fixed surrogate dictates the distribution used for LocalZO, and vice-versa. For example, appendix A.1 (eqn. 11, 12, 13) gives the surrogates obtained from different distributions of z, such as, Gaussian, Laplace and Uniform. Also, explicit equations governing the passage from the surrogate to the distribution is supplied in appendix A.2 ( eqn. 22, 23), where we calculated the PDFs for the most commonly used surrogate functions, such as, Sigmoid and FastSigmoid.

---

> > > > > ### Comment · Reviewer_DTqo · 2023-08-19
> > > > >
> > > > > Thank you for your detailed response. In summary, I still do not think the algorithm itself has much novelty. The theoretical analysis that the method can simulate arbitrary surrogate is somewhat interesting. However, the fact that the method lacks adaptivity during the training somewhat weaken the theory.
> > > > >
> > > > > Therefore, I would like to increase my score from 3 to 4.

---

> > > > > > ### Author Response · Authors · 2023-08-20
> > > > > >
> > > > > > Dear reviewer,
> > > > > >
> > > > > > In light of your latest comments, we believe there has been perhaps a misunderstanding of what we mean by "adaptive".
> > > > > >
> > > > > > On our side, by "adaptive" we meant that LocalZO can simulate arbitrary surrogate functions
> > > > > > (under some mild conditions explicit in the definition of surrogate function in our paper, e.g., theorem 3 assumes, that the integral to compute c exists).
> > > > > >
> > > > > > If by mentioning "how to adaptively select the distribution z during training, especially there is no closed form for the expected surrogate function" your concern is how we can sample from $\lambda$, when assumption on c does not hold
> > > > > > -- we can attempt to handle such situations with existing literature to sample un-normalized univariate distributions [1].
> > > > > >
> > > > > > It would be helpful if you could further clarify the term "adaptive"?
> > > > > >
> > > > > > [1] Bishop, Christopher (2006). "11:Sampling Methods". Pattern Recognition and Machine Learning

---

### Official Review · Reviewer_cVHq · 2023-07-07

**Soundness:** 3 good
**Presentation:** 3 good
**Contribution:** 3 good
**Rating:** 6
**Confidence:** 4

**Summary:**

This paper presented a direct SNN training algorithm that alleviates the loss of the gradient information and improves the performance of the SNN on multiple datasets, including both static image datasets and dynamic vision datasets.

**Strengths:**

- Rigorous theoretical and empirical analysis to justify the necessity of the Zeroth Order technique.

- Comprehensive experimental results with improved performance against the previous SoTA method.

**Weaknesses:**

- The proposed method is only verified on small-scale datasets such as CIFAR-10/100 and DVS-CIFAR-10. Since the proposed method shows improved performance on simple vision tasks, it is necessary to further verify the performance on large-scale datasets such as ImageNet-1K or ImageNet-100.

- I understand that static CIFAR datasets are standard vision tasks in most of the recent direct SNN training methods. However, one of the major advantages of SNN is the ability to process the spatial temporal visual information which widely exists in the captures of the event-based sensor/camera. Solely verifying the proposed method with the DVS-CIFAR10 dataset is insufficient, it is very useful for the research community if the author can provide the performance with more DVS datasets (e.g., IBM gesture, NCARS, N-CalTech101), which was reported by [1].

[1] AEGNN: Asynchronous Event-based Graph Neural Networks, CVPR'22.

- I wonder if the proposed method is applicable to the SpikeFormer [2]?

[2] Spikformer: When Spiking Neural Network Meets Transformer, ICLR'23

**Questions:**

Please refer to the Weaknesses section.

**Limitations:**

The major limitation of the paper is the insufficient experimental results.

---

> ### Author Rebuttal · Authors · 2023-08-09
>
> **W1:** The proposed method is only verified on small-scale datasets such as CIFAR-10/100 and DVS-CIFAR-10. Since the proposed method shows improved performance on simple vision tasks, it is necessary to further verify the performance on large-scale datasets such as ImageNet-1K or ImageNet-100.
>
> **A1:** We perform experiments of ImageNet-100 dataset using an integrate and fire (IF) SEW-Resnet34 model [7] with ZerO initialization [5]. We compare LocalZO (m=5, $\delta$=0.5) with the corresponding Gaussian Surrogate, both implemented using TET loss, trained for 200 epochs. The experiments are performed with standard augmentation (RandomResizedCrop(224),
> RandomHorizontalFlip), **with** and **without** ImageNet Policy [8]. The obtained top-1 accuracy is comparable to the SOTA results [6], and exceeds it with the presence of ImageNet Policy. We do not observe any significant loss of accuracy when compared to the surrogate method.
>
>
> | Datasets  | Surrogate |	LocalZO | Surrogate+Aug |  LocalZO+Aug |
> | :------------- | :-------------: | :---------: | :------------------: | :------------------: |
> | Imagenet-100 |  78.38 |	**78.58** | **81.58** | **81.56** |
>
>
> **W2:** I understand that static CIFAR datasets are standard vision tasks in most of the recent direct SNN training methods. However, one of the major advantages of SNN is the ability to process the spatial temporal visual information which widely exists in the captures of the event-based sensor/camera. Solely verifying the proposed method with the DVS-CIFAR10 dataset is insufficient, it is very useful for the research community if the author can provide the performance with more DVS datasets (e.g., IBM gesture, NCARS, N-CalTech101), which was reported by [1].
>
> **A2:**  We consider datasets suggested by the reviewer and compare results of LocalZO vs. Surrogate, where LocalZO uses the standard Gaussian distribution to sample z (m=5, $\delta$=0.5), and the Surrogate method uses the corresponding Gaussian surrogate as obtained in eqn. (9). We perform the experiment with TET loss vs. plain cross-entropy loss accompanied with tDBN, following the experimental settings in the paper.
>
> The events of the neuromorphic datasets are collected into event frames of dimension (2 $\times$ H $\times$ W), and the number of frames (a.k.a. number of bins) are considered as the temporal dimension for SNN. We set the number of frames to 10, and they are resized to dimension ($2 \times 48 \times 48$) for all the neuromorphic datasets. It is the same pre-processing step we followed to obtain DVS-CIFAR-10 results.
>
> We perform the experiments under two settings, **with** and **without** data augmentation, using the VGGSNN architecture reported in the paper. Under data-augmentaion we use the standard augmentation technique of RandomCrop(48, padding=4) and RandomHorizontalFlip. We train the models from scratch for 200 epochs (batch-size 64 for DVS-Gesture, and 16 for N-Caltech and NCARS), while the other training hyper-parameters remains the same as previously reported in the paper .
>
> For DVS-Gesture dataset (a.k.a. IBM-Gesture) we obtain the top-1 accuracy 98.43\% which is comparable to the state-of-the-art accuracies (98\%) reported for this dataset [1]. In N-Caltech-100 dataset we obtain top-1 accuracy of 82.99\%, which is higher than the reported SOTA(81.7\%) that uses transfer learning on SEW-ResNet-34 model pre-trained with Imagenet [2]. For NCARS dataset we obtain top-1 accuracy 96.96\% which is again higher than SOTA of 94.5\% [1]
>
> The comparison with the corresponding Gaussian surrogate shows that the randomness introduced by LocalZO frequently helps the training to obtain better generalization performance. The better generalization by LocalZO holds irrespective to the randomness introduced by data-augmentation, and irrespective to the accompanying tDBN or TET method.
>
> We thank the reviewer for suggesting DVS datasets, we shall include these results in the future version of the paper.
>
>
> | Datasets | Loss | Surrogate |	LocalZO | Surrogate+Aug |  LocalZO+Aug |
> | :----------- |:------: | :-------------: | :---------: | :------------------: | :------------------: |
> |DVS-Gesture | TET  | 94.9     | **98.04**| 96.09 | **98.43** |
> |                      | tDBN | 87.89   | 95.31     | 92.97 | 91.41 |
> | N-Caltech-101 | TET | 67.24 | **79.86** | 76.04 | **82.99** |
> |                         | tDBN |  68.4 | 74.65      | 75.58 | 79.05 |
> | N-CARS |TET |  95.42 | **96.78**	 | 95.09 | **96.96**|
> |                |tDBN | 94.06	| 95.96  | 94.83 | 95.68 |
>
> **W3:**  I wonder if the proposed method is applicable to the SpikeFormer [2] ?
>
> **A3:** Spikeformer uses derivative of the sigmoid function as the surrogate gradient for the Heaviside function (see Appendix C1 of [2]). On the other side, LocalZO method is able to simulate the derivative of the sigmoid function (see Appendix A.2.1 of our paper), and as such it is applicable to the Spikeformer. Although it would be a very interesting application of our method that we are happy to test in the future, it goes beyond of the present scope of our paper.
>
> [1] AEGNN: Asynchronous Event-based Graph Neural Networks, CVPR'22.
>
> [2] Spikformer: When Spiking Neural Network Meets Transformer, ICLR'23
>
> [3] "Sequence approximation using feedforward spiking neural network for spatiotemporal learning: Theory and optimization methods." ICLR'22.
>
> [4] "End-to-end learning of representations for asynchronous event-based data." ICCV'19.
>
> [5] “ZerO Initialization: Initializing Neural Networks with only Zeros and Ones” , TMLR'22
>
> [6] "Sharing Leaky-Integrate-and-Fire Neurons for Memory-Efficient Spiking Neural Networks." arXiv:2305.18360 (2023).
>
> [7] "Deep residual learning in spiking neural networks." Neurips (2021)
>
> [8] “AutoAugment: Learning Augmentation Policies from Data” arXiv:1805.09501 (2019)

---

> > ### Comment · Reviewer_cVHq · 2023-08-18
> > **Well-received rebuttal**
> >
> > The additional experimental results presented by the author further prove the performance of the proposed method. It will be great if the author could include these comprehensive experimental results in the next version of the paper.
> >
> > To that end, I will increase my score from 5 to 6.

---

### Decision · Program_Chairs · 2023-09-21

**Decision:**

Accept (poster)

**Comment:**

A zeroth-order method for training Spiking Neural Network is proposed. Three reviewers voted that the paper be accepted, but one of them DTqo recommends a reject. While all the reviewers appreciated the method, some had concerns about experiments that were resolved in the rebuttal. DTqo had concerns about novelty and lack of adaptivity. However, novelty is subjective and the method improves the result. Authors responded to the concern about adaptivity, but the reviewer didn't respond even after intervention from AC. I personally don't think its a reason enough to reject the paper. Therefore, I recommend accepting this paper.